# Multiple myeloma long-term survivors exhibit sustained immune alterations decades after first-line therapy

The long-term consequences of cancer and its therapy on the patients' immune system years after cancer-free survival remain poorly understood. Here, we present an in-depth characterization of the bone marrow immune ecosystem of multiple myeloma long-term survivors, from initial diagnosis up to 17 years following a single therapy line and cancer-free survival. Using comparative single-cell analyses combined with molecular, genomic, and functional approaches, we demonstrate that multiple myeloma long-term survivors exhibit pronounced alterations in their bone marrow micro-environment associated with impaired immunity. These immunological alterations were frequently linked to an inflammatory immune circuit fueled by the long-term persistence or resurgence of residual myeloma cells. Notably, even in the complete absence of any detectable residual disease for decades, sustained changes in the immune system were observed, suggesting an irre-versible 'immunological scarring' caused by the initial exposure to the cancer and therapy. Collectively, our study provides key insights into the molecular and cellular bone marrow ecosystem of long-term survivors of multiple mye-loma, revealing both reversible and irreversible alterations in the immune compartment.

The immune system plays a key role in the prevention, develop-ment, and treatment of cancer. Powerful immune surveillance mechanisms constantly monitor tissues to remove potentially can-cerous cells. However, malignant tumors can evade immune control or even hijack immunological processes to propel tumor growth. Notably, the interaction between the tumor and the immune system induces bidirectional adaptations. Well-studied examples for immunological changes induced by continuous exposure to tumor cells include the exhaustion and dysfunction of T cells, as well as the suppressive polarization of myeloid immune cells, such as tumor-associated macrophages or myeloid-derived suppressor cells[1–5]. In infectious diseases, irreversible immune dysfunction has been described, long after the infection has been cleared, a phenomenon termed immunological scarring[6,7]. However, whether cancer or cancer treatment may cause similar long-term consequences on the

immune system years after cancer-free survival remains poorly understood.

Multiple Myeloma (MM) is a hematologic neoplasm characterized by the clonal proliferation of malignant plasma cells within the bone marrow (BM). MM provides a prime example of a disease that depends on the interplay with its tumor microenvironment[8,9]. Recent bulk and single-cell genomic efforts dissected the clonal complexity as well as clonal evolution patterns of MM from precursor stages to sympto-matic disease and upon refractory cancer after multiple therapy lines[10–12]. While transcriptional stability has been observed in the transition from precursor states to MM progression, more dynamic shifts within the transcriptome and clonal outgrowth occurred upon refractory cancer[13]. Besides the genomic evolution of myeloma cells, substantial changes in the immune and stromal cell composition have been described across the different MM disease stages, promoting an

✉ e-mail: b.brors@dkfz-heidelberg.de; hartmut.goldschmidt@med.uni-heidelberg.de; c.imbusch@dkfz-heidelberg.de; michael.hundemer@med.uni-heidelberg.de; simon.haas@bih-charite.de

inflammatory BM microenvironment upon disease progression[8]. Cell-cell interactions within the BM appear to be crucial to mediate tumor growth in MM, highlighting the importance for a deeper understanding of the tumor ecosystem at different disease stages[14]. While recent studies on the MM ecosystem focused on disease progression from precursor stages as well as refractory disease, it remains unclear whether myeloma and myeloma therapy causes long-term alterations of the immune system years to decades after progression-free survival.

Despite improved therapy options, MM remains an incurable disease, and historically, only a minor fraction of MM patients experienced long-term survival (LTS) over 7 years after first-line therapy[15,16]. Nonetheless, even patients in complete remission (CR) without the detectable measurable residual disease (MRD) may ultimately experience biochemical progression years after progression-free survival. Previous studies on the LTS phenomenon in MM focused on quantitative changes in immune cell types[17–19]. However, the transcriptional evolution patterns of myeloma cells in LTS patients, as well as the long-term molecular adaptations of the BM microenvironment years after progression-free survival, remain unexplored.

Here, we characterize the BM immune ecosystem of MM long-term survivors at initial diagnosis (ID) and 7–17 years after first-line therapy with a standard induction regimen and high-dose therapy followed by autologous stem cell transplantation. These patients experienced long-term remission in the absence of any maintenance therapy for a median of 9 years prior to sampling (Supplementary Data 1). Since treatment regimens have changed in recent years and patients now receive permanent maintenance therapy, the selected patient cohort represents an ideal setting to study the immediate and long-term impact of cancer and cancer therapy on the immune system. Notably, our data demonstrate that LTS patients display sustained alterations in the immune microenvironment compared to age-matched controls. These changes are associated with the resurgence of disease activity but are also detectable in patients considered functionally cured, suggesting both reversible and irreversible long-term consequences of the disease and therapy. We identify bone marrow infiltrating inflammatory T cells as part of an inflammatory circuit, driving these sustained immune aberrations. Importantly, this disease-associated immune cell trafficking can be used to reliably track the re-initiation of the disease.

## Results

### The bone marrow immune ecosystem of multiple myeloma long-term survivor patients

The long-term alterations of the immune system years to decades after successful cancer therapy and LTS remain unknown. To elucidate the bone marrow immune ecosystem of LTS cancer patients, our study included 24 multiple myeloma patients who experienced LTS for 7 to 17 years (median 10.5 years) after first-line therapy with standard induction regimen and high dose therapy followed by autologous stem cell transplantation (Fig. 1a and Supplementary Data 1). Notably, the favorable outcome of these patients could not have been predicted by state-of-the-art risk stratification tools, as 10 out of 24 patients displayed an intermediate or poor prognosis according to the International Staging System (ISS)[20], and 4 patients even harbored high-risk cytogenetic aberrations. Average myeloma cell infiltration within the BM across all patients at ID was remarkably high (mean 50%). For 11 of these MM patients with paired longitudinal samples at ID and upon LTS 7–17 years post-diagnosis, we performed droplet-based single-cell RNA-sequencing (scRNAseq) of total BM mononuclear cells. In addition, CD3 + T cells were separately profiled in all cases by scRNAseq to ensure sufficient coverage of the T cell compartment, even in the presence of a high tumor burden. Bone marrow samples from three healthy, age-matched donors were included as controls, applying the identical workflow (Fig. 1a, Supplementary Fig. 1a, e and Supplementary Data 2). Following data integration, clustering and dimensionality

reduction across experiments, we analyzed 213,200 high-quality BM cells covering the vast majority of previously described hematopoietic cell types and cell states of the BM (Fig. 1b and Supplementary Fig. 1b). These included plasma cells, all hematopoietic stem and progenitor cell stages, T cell and natural killer (NK) cell populations, several dendritic cell and monocyte subpopulations as well as the main B cell differentiation states.

Comparing immune cell compositions of healthy donors with patients at ID revealed an expected enrichment for plasma cells and a trend towards higher amounts of cDC1 and NK cells, as well as a depletion of different B cell stages as described by previous studies (Fig. 1c, d and Supplementary Fig. 1c, d)[10,21]. At the LTS timepoint, the BM composition was partially normalized, however, a significant enrichment of the dendritic cell compartments cDC1 and cDC2 constituted a specific feature of LTS patients (Supplementary Fig. 1d). Besides changes in the BM cell type composition, we also observed considerable transcriptional perturbations within many BM-resident cell types, reflecting disease-associated adaptations of cellular transcriptomic states (Fig. 1c). To quantify these changes in cellular states associated with ID and LTS, we made use of DA-seq, a computational tool that measures how much a cell's neighborhood is dominated by a certain biological state (see "Methods"). As expected, a major transcriptomic remodeling from healthy to malignant plasma cells was observed at ID (Fig. 1e, f). In addition, significant transcriptomic changes occurred within CD14 + monocytes, CD16+ monocytes as well as T and NK cells. Importantly, while the transcriptomic remodeling of immune cells partially normalized during LTS, sustained signs of immune remodeling were maintained even decades after a single therapy line and in the absence of maintenance therapy for at least 4 years (median 9 years) (Fig. 1g and Supplementary Data 1).

### Malignant plasma cells may persist during long-term survival and display a transcriptionally stable phenotype

Recent studies reported dynamic transcriptional shifts of malignant plasma cells and clonal outgrowth during disease courses induced by therapeutic interventions[13]. However, it remains poorly understood whether plasma cells driving relapse years after tumor-free survival undergo molecular adaptations in the absence of therapy pressure. Moreover, it is unclear whether malignant plasma cells persist in the BM of LTS patients that are considered functionally cured.

To address these questions, we performed an in-depth analysis of plasma cells to explore the longitudinal changes of the tumor cell compartment throughout LTS. The transcriptional heterogeneity of the plasma cell compartment was reflected by patient-specific MM cell clusters and a cluster of putative healthy plasma cells to which all patients and the healthy controls contributed (Fig. 2a). Patient-specific clusters showed distinct gene expression patterns in line with published bulk RNA gene expression signatures, highlighting the diversity of our patient cohort (Supplementary Fig. 2a)[22]. As expected, the expanded plasma cell compartment at ID partially normalized upon LTS. However, some patients still harbored a high fraction of plasma cells at the LTS state (Fig. 2b). To delineate healthy and malignant plasma cells, we analyzed copy number aberrations (CNA) using inferCNV (see methods, Supplementary Fig. 3). Overall, 59 out of 63 CNAs detected by cytogenetics could also be identified by our single-cell analyses, permitting a clear discrimination between healthy and malignant plasma cells (Fig. 2c and Supplementary Fig. 2b, c). Furthermore, plasma cells classified as malignant almost exclusively expressed a single immunoglobulin light chain, whereas plasma cells classified as healthy contained both kappa and lambda-expressing cells, confirming the accuracy of our CNA analyses (Fig. 2d, e and Supplementary Fig. 2e). The fraction of malignant plasma cells within the overall plasma cell pool (termed 'malignancy score') was increased in LTS patients that had experienced a biochemical progression from

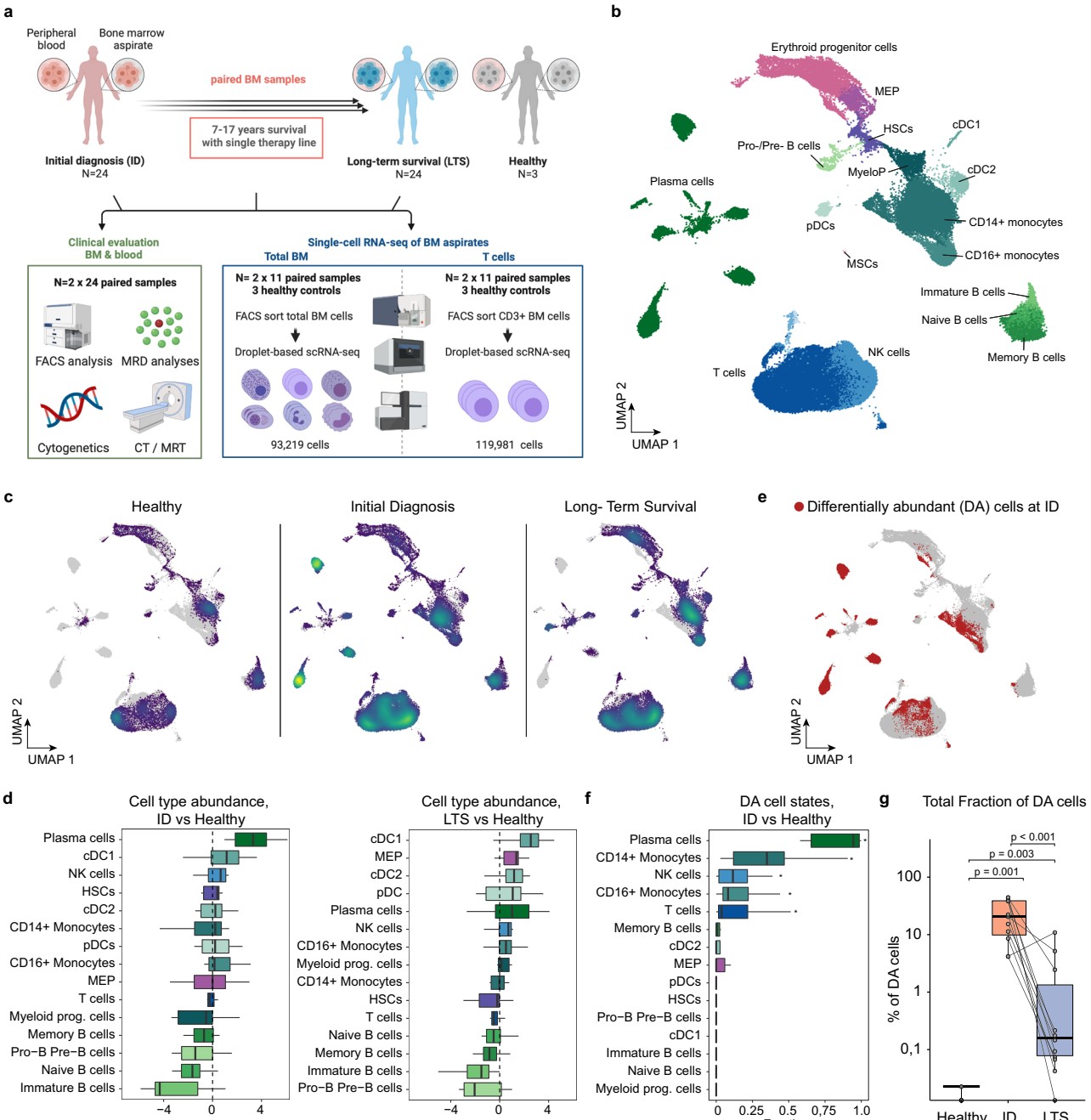

**Fig. 1 | The bone marrow immune ecosystem of multiple myeloma long-term survivor patients. a** Overview of the study design and experimental layout; created with BioRender.com. **b** Global UMAP representation of scRNAseq data of paired human BM samples from 11 MM long-term survivor patients at initial diagnosis (ID) and long-term survival (LTS), as well as 3 healthy, age-matched controls. **c** Global UMAP split by clinical groups. The density and distribution of cells are color-coded. Gray represents all remaining cells. **d** Changes in cell type abundancies between ID ($n = 11$) or LTS ($n = 11$) in comparison to healthy donors ($n = 3$). **e** Global UMAP highlighting differentially abundant cells (red) determined by DA-Seq at initial diagnosis as compared to cells from healthy controls. **f** Fractions of differentially abundant (DA) cells compared to all cells per cell type and patient at initial diagnosis ($n = 11$). Benjamini-Hochberg (BH) adjusted significant differences

($p < 0.05$) evaluated by unpaired two-sided Wilcoxon rank sum test are highlighted. **g** Fractions of DA cells compared to all cells per patient within ID, LTS or healthy controls (Healthy). Dots represent sample means. BH corrected $p$-values from unpaired (Healthy/ID, Healthy/LTS) and paired (ID/LTS) two-sided Wilcoxon rank-sum tests are shown. If not stated otherwise, paired human BM samples from 11 MM patients at ID and LTS, as well as 3 healthy, age-matched controls, were used for comparison. Abbreviations: HSCs: hematopoietic stem cells, MEP: megakaryocyte-erythrocyte progenitors, MyeloP: myeloid progenitors, cDC1/2: conventional dendritic cells 1/2, pDCs: plasmacytoid dendritic cells, NK: natural killer cells, MSCs: mesenchymal stromal cells; ID: initial diagnosis, LTS: long-term survival. Box plots: center line, median; box limits, first and third quartile; whiskers, smallest/largest value no further than 1.5*IQR from the corresponding hinge.

complete remission (CR) after a long-term remission phase, hereafter termed non-CR patients (Fig. 2f and Supplementary Fig. 2d). As expected, patients that were in clinical CR harbored less or no malignant plasma cells. Moreover, the fraction of malignant cells defined by

CNAs correlated with the result obtained from next-generation flow cytometry for the detection of measurable residual disease (MRD) (Fig. 2g). The mapping of CNAs in the single-cell data of the plasma cell compartment enabled us to address the question of how myeloma

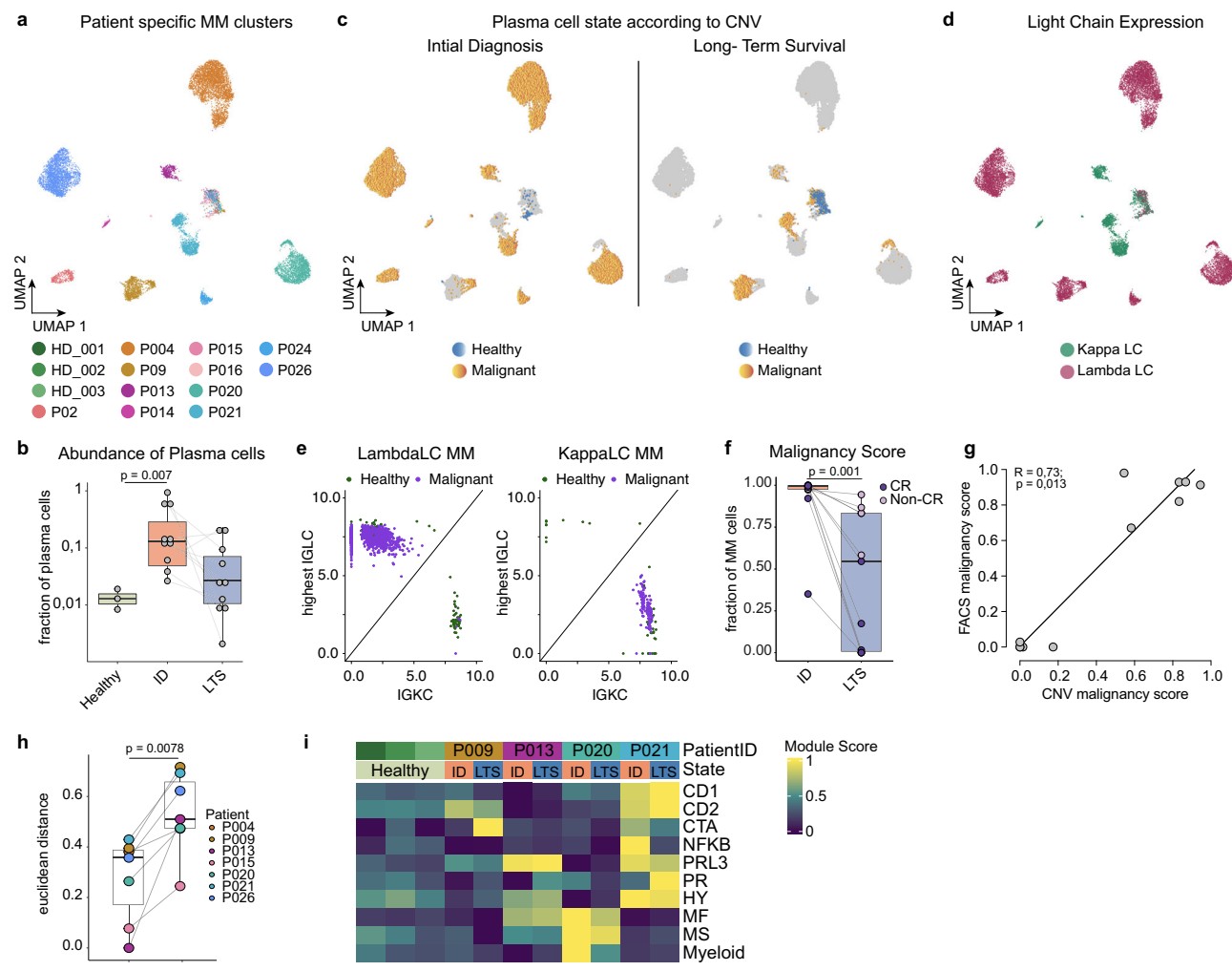

**Fig. 2 | Malignant plasma cells frequently persist during long-term survival and display a stable transcriptional phenotype. a** UMAP embedding of the BM plasma cell (PC) compartment colored by the patient. **b** PC fraction of total BM cells summarized by the patient and compared between clinical groups (Healthy (*n* = 3), ID (*n* = 10), LTS (*n* = 10)). Dots indicate the PC fraction of total BM cells for each sample. Significance was tested by a two-sided unpaired Wilcoxon rank sum test. **c** Split UMAP of PCs by clinical groups (ID, LTS) highlighting their malignancy annotation (healthy, malignant) derived from inferCNV. The remaining cells are grayed out. **d** PC UMAP highlighting the dominant immunoglobulin light chain expression. **e** Representative scatter plots of immunoglobulin expression (highest lambda chain (IGLC) and kappa chain (IGKC)) of healthy (green) and malignant (violet) PCs. **f** Malignancy score (malignant PC fraction of total PCs) per patient at ID and LTS (*n* = 9). Large dots indicate the malignant PC fraction of total BM cells for each sample. Significance was tested by a two-sided paired Wilcoxon signed rank test. **g** Correlation of malignancy score from flow cytometry MRD (number of Light

Chain restricted plasma cells/all plasma cells) with malignancy score from inferCNV analysis (number of malignant cells/all plasma cells). Spearman's Rho and the significance level of correlation are indicated using a two-sided test based on the t distribution. **h** Euclidean distance of malignant plasma cells between ID and LTS within patients (*n* = 7) compared to the Euclidean distance of malignant plasma cells at ID and respective nearest neighbor. Patients with less than 2 cells per clinical state were excluded. Significance was tested by a one-sided paired Wilcoxon signed rank test. **i** Heatmap showing average expression patterns (module scores; scaled by row) of known bulk RNA signatures (Broyl et al. 2010) per patient and clinical state. If not stated otherwise, paired human BM samples from 11 MM patients at ID and LTS, as well as 3 healthy, age-matched controls, were used for comparison. Abbreviations: PC: plasma cells; ID: initial diagnosis; LTS: long-term survival; IGLC: immunoglobulin light chain; LC: lambda chain; KC: kappa chain. Box plots: center line, median; box limits, first and third quartile; whiskers, smallest/largest value no further than 1.5*IQR from the corresponding hinge.

cells develop throughout the LTS state upon recurring disease activity. Malignant myeloma cells from the same patient at ID and LTS shared the highest transcriptional similarity to each other in comparison to myeloma cells from other patients (Fig. 2c, h). This suggested a high transcriptional stability of plasma cells upon resurgence of disease activity even after long-lasting remission over years to decades. However, minor adaptations in the transcriptomic makeup between matched malignant plasma cells at ID and LTS were observed, as indicated by minor, but specific changes in the UMAP representation (Fig. 2c). To further study the molecular adaptations of myeloma cells, we focused on 4 patients with sufficient malignant cells captured for both

matching clinical states to reliably obtain the subclonal composition of the respective patients (Supplementary Fig. 3). Notably, we observed a changing subclonal composition which translated into specific changes of gene expression patterns of published transcriptomic signatures that are commonly used to categorize transcriptional patterns of myeloma cells (Fig. 2i)[22]. For example, P009 gained a cancer-testis antigen (CTA) expression pattern, which is reported to be associated with a proliferative myeloma disease, whereas P021 lost the previously expressed NFKB signature upon the resurgence of disease (Fig. 2i). Together, our observations demonstrate that malignant plasma cells may persist in LTS patients and display an overall transcriptionally

stable phenotype that is maintained for decades, while specific transcriptomic adaptations may occur.

**Multiple myeloma long-term survivor patients display sustained signs of immune remodeling decades after a single therapy line**
While specific compositional changes in the BM microenvironment of LTS patients have been reported[17–19], it remains unknown whether these cell types adopt a cellular state similar to healthy BM cells or maintain signs of their current or past exposure to malignant plasma

cells or therapy. Our initial analyses revealed a major transcriptomic remodeling of BM-resident immune cells during the disease course, with monocytic, T, and NK cell compartments displaying the most extensive alterations in cell states besides the plasma cell compartment (Fig. 1f). To further investigate these molecular changes across the clinical states, we first focused on the most remodeled cell compartment, classical CD14+ monocytic cells (Fig. 3a). In line with our global DA-seq analysis, the majority of monocytes from ID patients clustered separately from monocytes of healthy donors, reflecting a

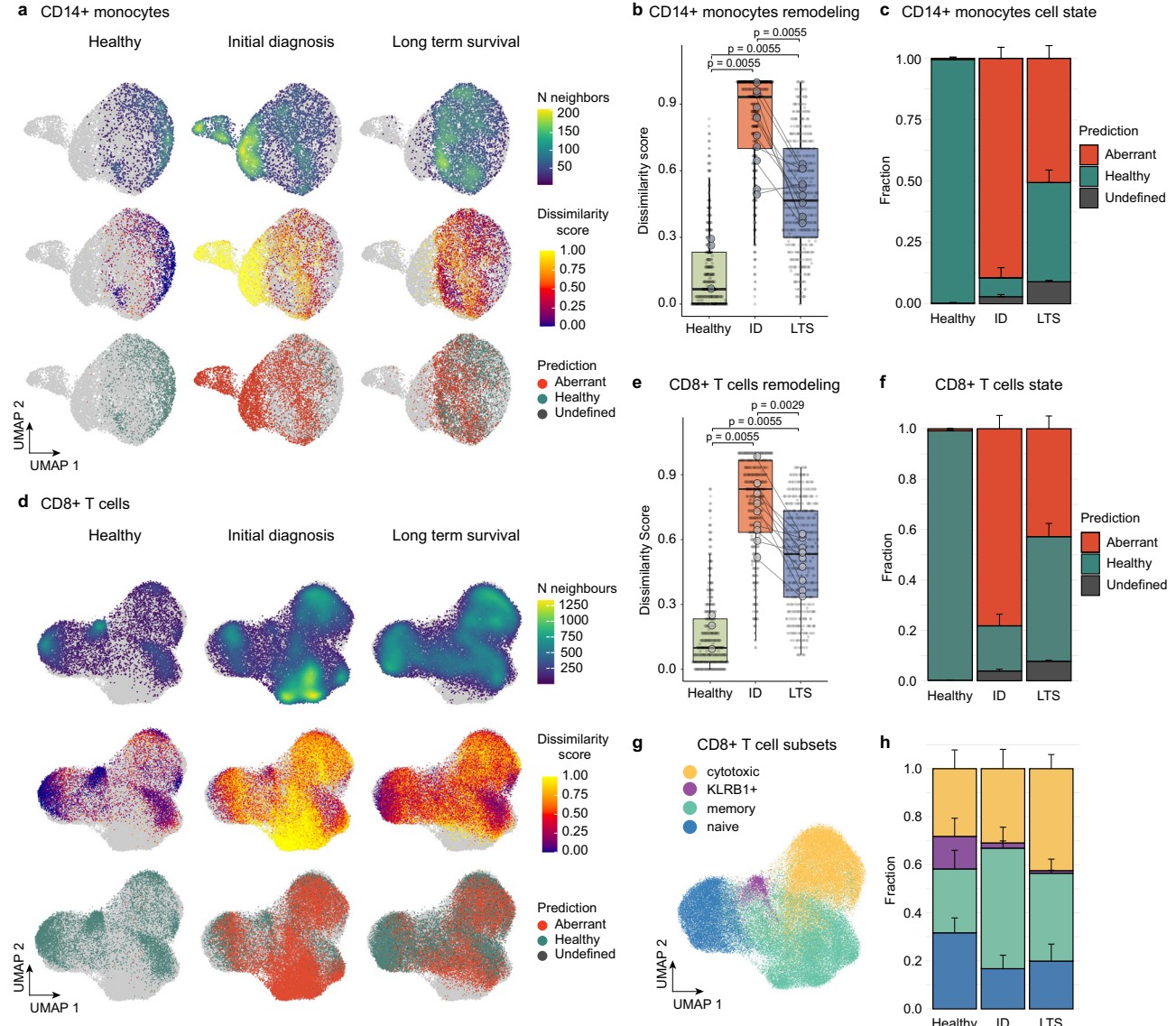

**Fig. 3 | Multiple myeloma long-term survivor patients display sustained signs of immune remodeling decades after a single therapy line. a** UMAP of CD14 + monocytes from the BM dataset split by clinical groups. Cells are colored by the density (top row), dissimilarity score (middle row), and dissimilarity-based classification into aberrant-like, healthy-like, and undefined cell states (bottom row). Cells from the respective other clinical states are depicted in gray. **b** Distribution of the dissimilarity score by clinical group summarizing the remodeling of CD14 + monocytes. Large dots indicate sample means. Benjamini-Hochberg adjusted *p*-values from unpaired (Healthy(*n* = 3)/ID(*n* = 11), Healthy (*n* = 3)/LTS(*n* = 11)) and paired (ID(*n* = 11)/LTS(*n* = 11)) two-sided Wilcoxon rank-sum tests are shown. **c** Bar plot summarizing fractions of predicted cell states by the clinical group from (**a**). **d** UMAP of CD8 + T cells split by clinical groups. Cells are colored by the density (top row), dissimilarity score (middle row), and dissimilarity-based classification into aberrant-like, healthy-like, and undefined cell states (bottom row). Cells from

the corresponding other clinical states are shown in a grayscale. **e** Distribution of the dissimilarity score by clinical group summarizing the remodeling of CD8 + T cells. Large dots indicate sample means. Benjamini-Hochberg adjusted *p*-values from unpaired (Healthy(*n* = 3)/ID(*n* = 11), Healthy(*n* = 3)/LTS(*n* = 11)) and paired (ID(*n* = 11)/LTS(*n* = 11)) two-sided Wilcoxon rank-sum tests are shown. **f** Bar plot summarizing fractions of predicted cell states by the clinical group from (**d**). **g** UMAP of CD8 + T cells, classified into naïve, memory, cytotoxic, and KLRB1 + subsets. **h** Bar plot summarizing fractions of cell subsets by the clinical group from (**g**). If not stated otherwise, paired human BM samples from 11 MM patients at ID and LTS, as well as 3 healthy, age-matched controls, were used for comparison. Abbreviations: BM: bone marrow; ID: initial diagnosis; LTS: long-term survival. Bar plots: Error bars indicate the standard error of the mean (SEM); Box plots: center line, median; box limits, first and third quartile; whiskers, smallest/largest value no further than 1.5*IQR from the corresponding hinge.

disease-associated transcriptomic remodeling. Notably, this remodeling partially normalized in the LTS state, although a considerable number of monocytes maintained a remodeled state years to decades after a single therapy line (Fig. 3a). To quantify the transcriptionally perturbed cells in the diseased states, we introduced a 'dissimilarity score' approximating whether a cell's neighborhood is dominated either by the healthy or the disease state. Combining the dissimilarity score with machine learning-based approaches enabled us to classify cells as 'healthy-like' or 'aberrant-like' with high accuracy and a low false prediction rate (see methods). These analyses revealed that classical monocytes from patients at ID showed a high degree of dissimilarity to healthy monocytes and were frequently classified as 'aberrant-like'. Upon LTS, only a partial normalization was observed, suggesting a sustained transcriptional remodeling throughout LTS in a subset of monocytes (Fig. 3b, c).

To investigate whether other immune cell types also display sustained transcriptional changes in the LTS state, we next focused on the T cell compartment. CD8 + T cell states were annotated in naive, memory, effector, as well as KLRB1 + cells based on known transcriptomic marker genes (Fig. 3g and Supplementary Fig. 4a–c). Notably, also in the CD8 + T cell compartment, a sustained transcriptional remodeling was observed upon long-term survival (Fig. 3d–f). Moreover, a significant and irreversible depletion of KLRB1 + CD8 + T cells was observed at the ID state and maintained throughout LTS (Fig. 3h).

In line with our observations from the classical monocyte and CD8 + T cell compartments, we observed a remodeling of non-classical CD16 + monocytes, as well as the CD4 + T and NK cell states at ID, which was partially sustained throughout LTS (Supplementary Fig. 4d–o). Together, our data reveals a major transcriptional remodeling across several cell types of the bone marrow immune microenvironment during active MM disease, which is sustained in a subset of cells throughout long-term survival.

## An inflammatory circuit is associated with immune remodeling during active disease and long-term survival

To characterize disease-associated molecular programs responsible for the acute remodeling in the bone marrow immune ecosystem at ID, we performed a comprehensive gene set enrichment analysis (GSEA) comparing aberrant-like cell states with cells from healthy controls within all cell types of the bone marrow that displayed disease-associated remodeling. This analysis revealed a globally upregulated inflammatory program (Hallmark TNFA signaling via NFKB and Hallmark inflammatory response) shared across all remodeled BM cell types, as well as cell type-specific changes (Fig. 4a). In particular, aberrant monocytes acquired a pro-inflammatory phenotype. The expression of inflammatory genes in monocytes correlated with their dissimilarity to healthy monocytes, peaked in ID patients, and partially reversed throughout LTS (Fig. 4b). However, the remaining 'aberrant-like' monocytes in the LTS state specifically displayed a sustained inflammatory phenotype, suggesting a persistent inflammatory response of the classical monocyte compartment even decades after the first line therapy (Fig. 4c). As part of the inflammatory response, 'aberrant-like' monocytes displayed an increased chemokine activity and expressed increased levels of proinflammatory cytokines and chemokines, including *CCL3, IL1B* and *CXCL8*, with the latter two known to support myeloma cell growth and survival (Fig. 4d, e and Supplementary Fig. 5a–e)[23]. Interestingly, the corresponding receptors of *CXCL8, CXCR1,* and *CXCR2* were mainly expressed on NK cells, suggesting a role for *CXCL8* in the regulation and induction of leukocyte migration as reported previously (Supplementary Fig. 5f). NK cells themselves switched from a cytotoxic to an inflammatory phenotype with increased chemokine activity, which was maintained throughout the LTS state (Fig. 4a, f).

To explore the interaction network between plasma cells and their microenvironmental cells at ID, we used CellPhoneDB[24] to infer intercellular communications (see "Methods"). We observed the highest number of interactions between myeloid and plasma cells (Fig. 4g). Notably, these interactions were significantly increased between remodeled CD14 + monocytes and plasma cells, suggesting that the remodeled state of CD14 + monocytes may be mediated by the interaction with plasma cells (Supplementary Fig. 5g).

Importantly, remodeled T and NK cells were the main producers of the proinflammatory master cytokine interferon-gamma (*IFNG*) both at ID and LTS (Fig. 4h, i and Supplementary Fig. 5h). Moreover, remodeled T and NK cells displayed significantly increased expression of the inflammatory chemokines *CCL3, CCL4* and *CCL5*, suggesting that they act as major regulators of the observed acute and sustained BM inflammation (Supplementary Fig. 5i–k). In line with an increased synthesis of proinflammatory cytokines by aberrant lymphocytes, including *IFNG*, we observed the strongest *IFNG* response in aberrant myeloid cells, including CD14 + and CD16 + monocytes as well as cDC2s (Fig. 4a). Notably, the interferon-inducible chemokine *CXCL10 was* mainly expressed by CD16 + monocytes peaking at ID and being maintained at a lower level throughout LTS (Supplementary Fig. 5l, m). Aberrant *IFNG*-expressing CD8 + T cells and NK cells specifically expressed *CXCR3*, the chemokine receptor mediating migration towards *CXCL10* sources, which we will elucidate in detail in the next section (Fig. 4j and Supplementary Fig. 5n).

In summary, these data suggest that upon MM disease activity in the BM, inflammatory signals drive a positive feedback loop with IFNG secretion by aberrant lymphocytes inducing the release of *CXCL10* from myeloid cells. This in turn may lead to the recruitment of *CXCR3* + inflammatory CD8 + T cells to the BM (see below) causing an inflammatory circuit which is maintained at a lower level in LTS patients (Supplementary Fig. 6a).

## Bone marrow infiltration of inflammatory T cells is associated with myeloma burden and serves as an accessible biomarker for disease activity

To characterize the origin and phenotype of disease-associated remodeled immune populations, we focused on aberrant CD8 + T cells as key producers of inflammatory cytokines throughout ID and LTS. Gene expression analyses of the scRNAseq data revealed the chemokine receptor *CXCR3* and the amino acid transporter *LAT1* as accurate biomarkers for a disease-associated inflammatory CD8 + T cell state (Fig. 5a, b). To further assess the value of surface *CXCR3* expression as a marker for myeloma-associated CD8 + T cells, we subjected BM *CXCR3* + and *CXCR3*- CD8 + T cells from an independent cohort of 7 MM patients to bulk RNA-sequencing (Supplementary Fig. 6b). Importantly, scRNAseq-derived *CXCR3* expression was highly overlapping with both, the single-cell-derived gene signature defining aberrant CD8 + T cells (Fig. 5c) and the bulk RNA-sequencing-derived gene signature for CXCR3 + T cells within the BM (Fig. 5d). This confirms the specificity of surface *CXCR3* as a biomarker for remodeled inflammatory T cells.

Next, we performed multiplex immunofluorescence stainings on BM biopsies and confirmed the co-expression of *CXCR3* and *LAT1* on CD8 + T cells in MM patients (Fig. 5e). Importantly, the mean expression intensities of *CXCR3,* as well as *LAT1* in CD8 + T cells were highly elevated in MM patients compared to B cell Non-Hodgkin lymphoma and MDS control cohorts, confirming the specific enrichment of aberrant inflammatory CD8 + T cells in MM (Fig. 5f and Supplementary Fig. 6c). Notably, the fraction of detected aberrant inflammatory CD8 + T cells positively correlated with the number of *MUM1* + plasma cells, suggesting a tumor-load dependent accumulation of aberrant inflammatory CD8 + T cells in the bone marrow at ID, with *LAT1* and *CXCR3* serving as accurate biomarkers (Fig. 5f and Supplementary Fig. 6c).

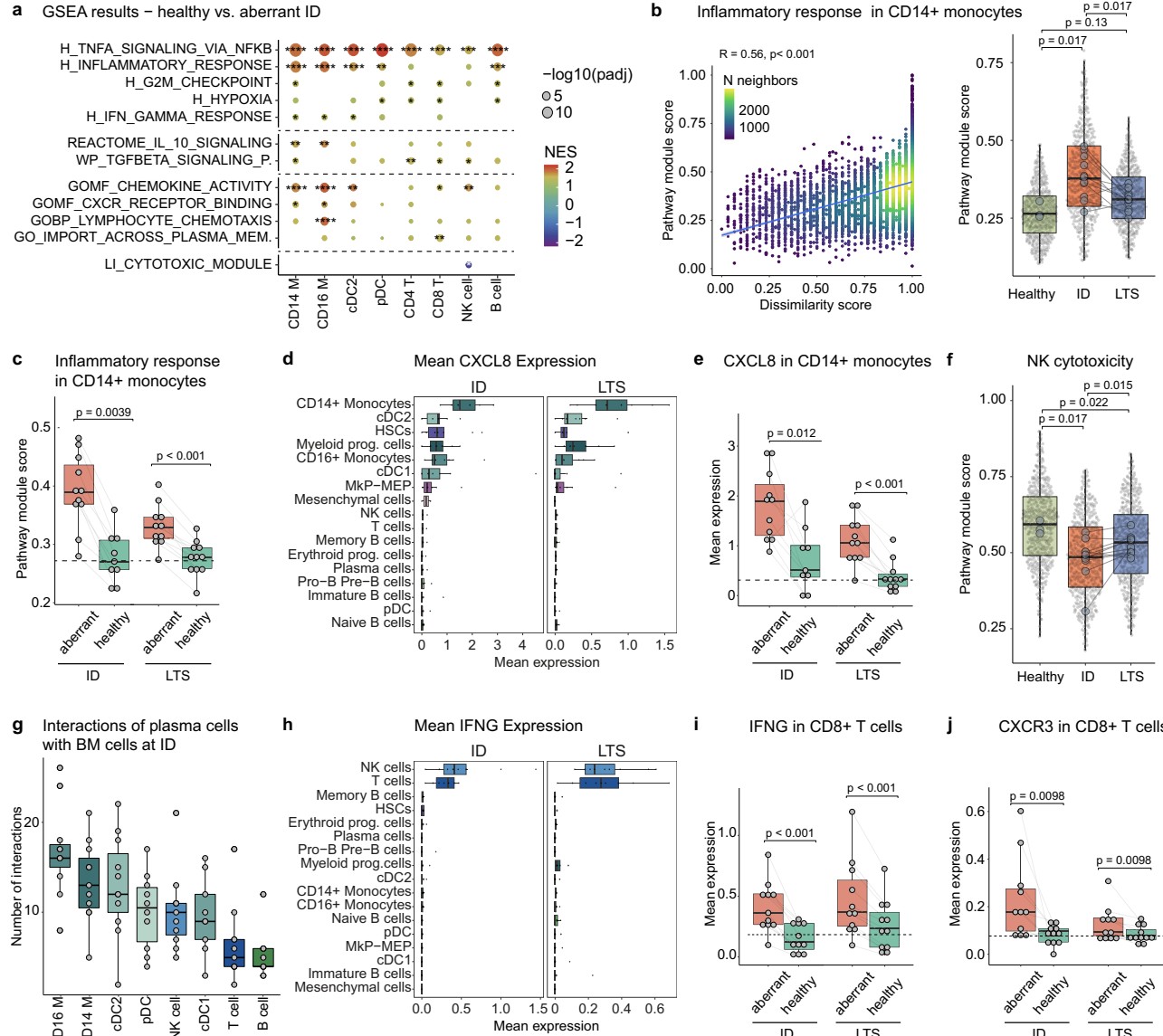

**Fig. 4 | An inflammatory circuit underlies immune remodeling during active disease and long-term survival. a** Significantly enriched gene sets in aberrant cell types at ID versus healthy controls. Selected gene sets are shown. Benjamini-Hochberg adjusted *p*-values are encoded by dot size, colors represent normalized enrichment scores (NES). Stars mark significant enrichment of the selected gene sets. **b** Left, correlation between indicated module score and dissimilarity score in CD14 + monocytes. Cell density is color-coded. Spearman's Rho and the significance level of correlation are indicated. Right, distribution of indicated module score by clinical group. Benjamini-Hochberg adjusted *p*-values from unpaired (Healthy/ID, Healthy/LTS) and paired (ID/LTS) two-sided Wilcoxon rank-sum tests are shown. **c** Boxplots of indicated module score (see b) in CD14 + monocytes split by clinical group and cell state prediction (n(ID/healthy) = 9, all other *n* = 11). The dashed line represents the mean expression of healthy controls. Significance was tested by paired two-sided Wilcoxon rank-sum tests. **d** Mean CXCL8 expression at ID and LTS per patient. **e** Boxplots of mean CXCL8 expression in CD14 + monocytes split by clinical group and cell state prediction. The dashed line represents the

mean expression of healthy controls. Significance was tested by paired two-sided Wilcoxon rank-sum tests. **f** NK cytotoxicity module score in NK cell subsets. Benjamini-Hochberg adjusted *p*-values from unpaired (Healthy/ID, Healthy/LTS) and paired (ID/LTS) two-sided Wilcoxon rank-sum tests are shown. **g** Predicted the number of interactions between plasma cells and immune cells at ID using Cell-PhoneDB. **h** Mean interferon-gamma (IFNG) expression at ID and LTS per patient. **i, j** Boxplots of IFNG (**i**) and CXCR3 (**j**) expression in CD8 + T cells split by clinical group and cell state prediction (n per group/condition = 11). The dashed line represents the mean expression of healthy controls. Significance was tested by paired two-sided Wilcoxon rank-sum tests. If not stated otherwise, paired human BM samples from 11 MM patients at ID, LTS, and 3 healthy controls were used for comparison. Abbreviations: CD14/CD16_M: CD14 + /CD16 + monocytes; cDC: conventional dendritic cells; pDC: plasmacytoid dendritic cells; CD4/CD8_T: CD4 + / CD8 + T cells; NK: natural killer cells. Box plots: center line, median; box limits, first and third quartile; whiskers, smallest/largest value no further than 1.5*IQR from the corresponding hinge.

To explore the origin of remodeled CD8 + T cells, we determined RNA velocities to predict the future cell state based on ratios of spliced to unspliced mRNAs (see methods). As reported in previous studies, this analysis revealed the transient and connected states of the main T cell subsets[25] (Fig. 5g). However, the cluster comprising aberrant inflammatory T cells, marked by *LAT1* and *CXCR3* expression and a high dissimilarity score, appeared disconnected to the cluster harboring the

main homeostatic BM-resident T cell subsets (Fig. 5g). As described above, *CXCR3* is a chemokine receptor mediating migration towards the chemoattractants *CXCL10*, which is synthesized at increased levels in the BM upon MM (Supplementary Fig. 5l, m). These observations point towards a chemokine-mediated infiltration of inflammatory T cells from the periphery to the BM. To further explore this, we quantified the *CXCR3* expression on CD8 + T cells of paired BM and

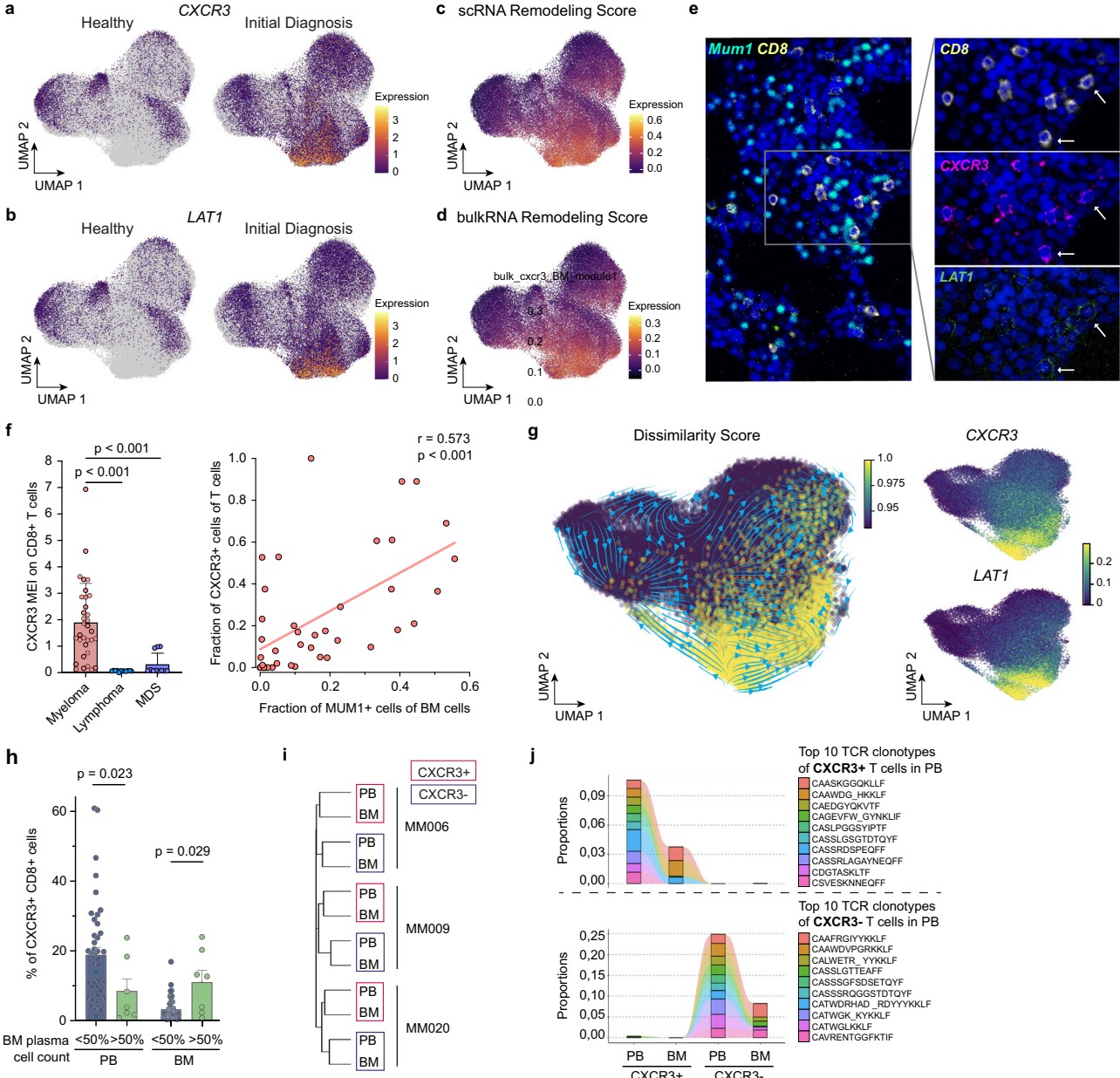

**Fig. 5 | Bone marrow infiltration of inflammatory T cells is associated with myeloma burden and serves as an accessible biomarker for disease activity.** **a–d** UMAP of CD8 + T cells with *CXCR3* (**a**) and *LAT1* (**b**) expression highlighted, split between healthy donors and ID patients; and highlighted 'scRNA Remodeling Score' (derived from scRNAseq data: healthy versus aberrant CD8 + T cells) (**c**) and 'bulkRNA Remodeling Score' (derived from RNAseq analysis of CXCR3 + versus CXCR3- CD8 + T cells) (**d**). **e** Multiplex immunofluorescence of *MUM1* (plasma cells), *CXCR3*, and *LAT1* in representative BM area of an MM patient (arrows indicated examples of *LAT1* and *CXCR3* co-expression on T cells). **f** Left, *CXCR3* mean expression intensity (MEI) on BM CD8 T cells of 33 MM, 12 B Non-Hodgkin lymphoma, and 11 MDS patients. Benjamini-Hochberg adjusted *p*-values from unpaired two-sided Wilcoxon rank-sum tests are shown; Right, Spearman correlation of fraction of CXCR3 + T cells (threshold: >10 T cells) with tumor burden (MUM1 + plasma cells in the BM) (*n* = 40). **g** UMAP of CD8 + T cells with highlighted velocities (arrows), dissimilarity score (yellow), and imputed *CXCR3* and *LAT1* expression.

**h** Fraction of CXCR3 + CD8 + T cells in PB (*n* = 50) and BM (*n* = 50) in patients with low/intermediate (< 50% plasma cells, blue) or high tumor burden (> 50% plasma cells, green). Significance was tested by a two-sided unpaired Wilcoxon rank sum test. **i** Hierarchical clustering CD8 + T cells (+/− CXCR3) from BM and PB (*n* = 3) by clonotypes of T cell receptor (TCR) repertoire using the Jaccard index of repertoire similarity. **j** Clonotype tracking by representative CDR3 amino acid sequence of shared clonotypes between the top 10 most abundant TCR clonotypes from CXCR3 + (top row) and CXCR3- (bottom row) PB CD8 + T cells across T cell subsets in PB and BM. Two representative patients are shown (see also Supplementary Fig. 6). Amino acid clonotype sequences are indicated. Bar plots: Error bars indicate the standard error of the mean (SEM). Abbreviations: FACS: fluorescence-activated cell sorting; BM: bone marrow; PB: peripheral blood; IF: immunofluorescence; MEI: mean expression intensity; MDS: myelodysplastic syndrome; ASCT: autologous stem cell transplantation; MFI: mean fluorescence intensity; TCR: T cell receptor.

peripheral blood (PB) samples from 50 MM patients via flow cytometry (Supplementary Fig. 6b). In line with our hypothesis, in patients with low tumor burden (< 50%), *CXCR3* + T cells were mainly present in the peripheral blood and not in the BM (Fig. 5h). In contrast, in patients with high tumor burden (> 50%) the number of *CXCR3* + T cells

decreased in PB, while an increased number of *CXCR3* + T cells was observed in the BM, suggesting a tumor-load dependent migration of inflammatory T cells to the BM.

To further validate this finding, we isolated bulk CD8 + T cells from LTS patients, as well as *CXCR3* + and *CXCR3*- CD8 + T cell subsets

from paired PB and BM samples of newly diagnosed MM patients, and performed RNA-sequencing, followed by mapping the T cell receptor (TCR) repertoire (Supplementary Fig. 6b, d). While we observed a trend towards a reduced clonal diversity in T cells of LTS patients, we did not observe any indication for clonal expansion of inflammatory *CXCR3* + CD8 + T cells if compared to their *CXCR3*- CD8 + T cell counterparts (Supplementary Fig. 6d). In contrast, hierarchical clustering based on TCR repertoire information revealed a striking overlap of the *CXCR3* + fractions from PB and BM for each patient, indicating a close relation between remodeled CD8 + T cells in the BM with *CXCR3* + CD8 + T cells in PB (Fig. 5i and Supplementary Fig. 6e). In line with this, the clonotypes of the top 10 clones in *CXCR3* + T cells from the PB showed a high overlap with the top clonotypes in *CXCR3* + T cells from BM fraction but not with their *CXCR3*- negative counterparts, suggesting a disease-associated infiltration of inflammatory T cells from the periphery to the BM (Fig. 5j and Supplementary Fig. 6f).

Together, our data suggest that upon MM disease activity, inflammatory CD8 + T cells are recruited to BM, where they serve as key players in the establishment and maintenance of the sustained inflammatory BM remodeling at ID and LTS (Supplementary Fig. 6a). BM infiltration by inflammatory T cells is associated with myeloma burden and serves as an accessible biomarker for disease activity that can be measured both in the BM and the peripheral blood.

## Immune remodeling in LTS patients is associated with future disease resurgence and impaired immune function even in the absence of measurable disease

The sustained immune alterations observed in LTS patients may be caused by the initial exposure to the cancer, therapeutic interventions, the persistence of residual MM cells, or a combination of these factors. To systematically dissect potential sources of sustained immune alterations, we first investigated the relationship between residual MM cells and immune perturbations in LTS patients. Specifically, we assessed the extent of immune cell remodeling in relation to the fraction of malignant plasma cells within the total plasma cell population, as indicated by the CNV malignancy score described above. Indeed, our analysis revealed that microenvironmental immune remodeling, as determined by DA-seq-based prediction scores, dissimilarity scores, and surrogate CXCR3 expression in CD8 + T cells, correlated with the proportion of malignant plasma cells in the bone marrow of LTS patients (Fig. 6a, b andSupplementary Fig. 7a).

Clinical follow-up allowed us to differentiate LTS patients with complete remission (CR) from those with remaining or resurgent MM cells (non-CR patients) and from patients that remained in CR for four years following sample collection for scRNAseq (termed sustained CR) (Supplementary Fig. 7b). At the time of sample collection, the ratio of BM to peripheral blood CXCR3 + CD8 + T cells increased progressively from healthy donors to sustained CR patients and those losing CR, to patients at initial diagnosis (ID), reflecting the respective disease burden across these clinical states (Fig. 6c). Notably, CR patients transitioning to non-CR status within the subsequent four years exhibited a significantly higher BM to blood CXCR3 + CD8 + T cell ratio compared to those maintaining in sustained CR (Fig. 6d). In line with an increased CD8 + T cell infiltration into the BM, an increased CD4 + to CD8 + T cell ratio in the blood was associated with future relapse from CR during LTS (Supplementary Fig. 7c). Collectively, these findings uncover persistent or resurgent MM cells as an important factor for sustained immune perturbation in the BM, and suggest that blood measurements, specifically CXCR3 expression on T cells, may be used as accessible biomarkers to track environmental perturbations associated with future relapse.

In line with the importance of residual MM cells for the sustained immune remodeling in LTS patients, the number of aberrantly classified immune cells gradually increased from healthy donors to CR and

non-CR patients (Fig. 6e, f and Supplementary Fig. 7d). However, even patients with sustained CR and no detectable disease activity at time of sample collection, as well as during an additional four-year follow-up period, exhibited a substantial presence of aberrantly classified T cells, monocytes, and NK cells (Fig. 6e–h and Supplementary Fig. 7d, e). This observation suggests the occurrence of long-lasting, irreversible immunological alterations that persist independently of residual MM cells. While the quantity of remodeled immune cells was correlated with residual disease activity, the degree of remodeling in aberrant immune cells was similarly pronounced in LTS patients with or without indications of remaining disease activity (Fig. 6g–k and Supplementary Fig. 7e–l). Moreover, compared to healthy controls, the naïve CD8 + T cell compartment of LTS patients showed a higher expression of an 'early T cell activation signature', even in the absence of any measurable disease activity, pointing towards a chronic pre-activatory state (Supplementary Fig. 7m)[26]. A gene within this 'early activation signature' was the well-studied surface marker CD69, expressed on activated T cells[27]. In line with our previous observations, CD8 + T cells from patients in CR and 'sustained' CR displayed increased CD69 surface protein levels when compared to healthy controls, providing additional evidence for a persistent long-term imprint in CD8 + T cells in the absence of disease activity (Supplementary Figs. 7n, 8c).

A recent study suggested clonal hematopoiesis (CH) as a potential driver of immune dysfunction in pediatric cancer survivors[28]. Targeted sequencing of BM samples revealed that 8 out 15 LTS patients carried mutations in genes related to CH (DNMT3A, $n = 5$, TP53, $n = 2$, ASXL1, $n = 1$, TET2 $n = 1$) (Supplementary Data 1). Although the prevalence of these mutations is higher compared to previously studied multiple myeloma (MM) cohorts[29,30], the small cohort size limits the ability to draw definitive conclusions. Notably, the degree of immune cell remodeling in LTS patients, both with and without clonal hematopoiesis of indeterminate potential (CHIP), was similarly extensive, suggesting that CHIP may not play a central role in the observed immune alterations of LTS patients (Supplementary Fig. 9).

To further delineate the effects of the initial tumor exposure from those resulting from therapy, we analyzed CXCR3 expression in CD8 + T cells as a surrogate marker for immunological disturbances in triple-matched samples from 136 MM patients collected at initial diagnosis after induction therapy and following high-dose melphalan treatment with autologous stem cell transplantation. Notably, the fraction of CXCR3-positive T cells showed a slight increase throughout the therapy, hinting at potential cumulative effects of the treatment (Fig. 6l). Yet, the fraction of CXCR3-positive T cells of MM patients measured at initial diagnosis (before treatment) was highly correlated with the fractions of CXCR3-positive T cells of the same patients measured throughout the different treatment phases (Fig. 6m; Wilcoxon signed rank tests < 0.001). This suggests that in addition to therapy-related effects, the initial exposure to the tumor, which plays a crucial role in determining immune remodeling at initial diagnosis (compare Fig. 5f and Supplementary Fig. 6c), likely also contributes to the immune alterations observed in LTS patients.

To assess whether sustained immune aberrations translate into changed T cell functionality, we measured the capacity of T cells from LTS patients to produce cytokines upon T cell activation. For this purpose, sorted CD3 + T cells were stimulated with PMA and Ionomycin, and intracellular cytokine production (TNFa, IFNG, IL2) was measured as a surrogate parameter for T cell functionality (Fig. 6n). Notably, stimulated T cells from LTS patients produced significant lower amounts of all measured cytokines compared to control samples from healthy donors and early-stage MM patients (Fig. 6o). Of note, impaired T cell functionality was also observed in patients with no measurable disease activity and sustained CR, suggesting a sustained immunological scarring in LTS patients. This aligns with broad

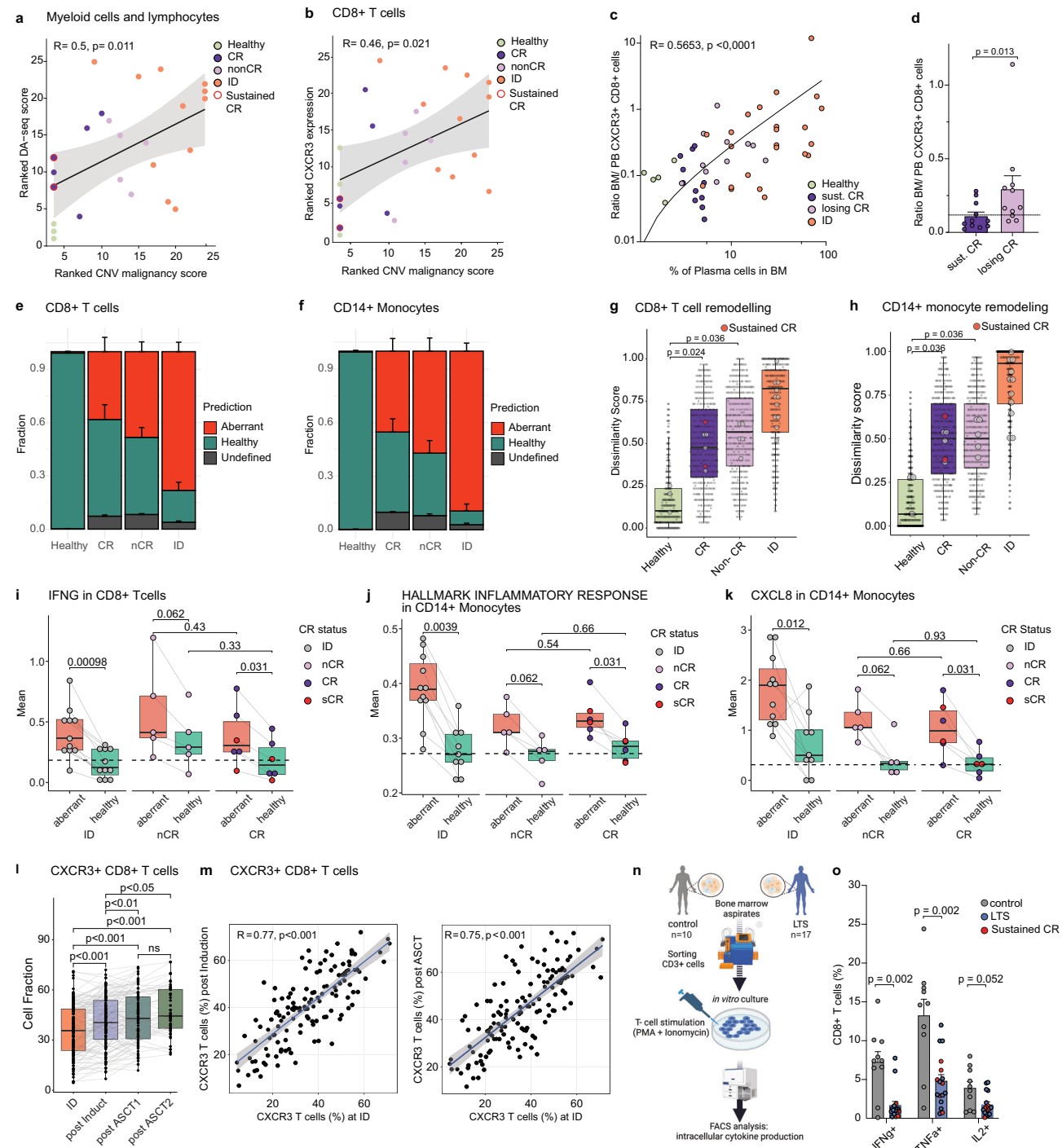

evidence suggesting that chronic exposure to inflammatory stimuli causes impaired T-cell functionality[31].

Collectively, these findings suggest that the immune alterations observed in LTS patients are likely not only due to the persistence or resurgence of malignant plasma cells but also result from long-lasting, irreversible immunological changes induced by the initial exposure to the disease and subsequent therapeutic interventions.

## Discussion

The long-term consequences of cancer and cancer therapy on the immune system remain poorly understood. In this study, we have comprehensively investigated the immune ecosystem in MM LTS patients, years to decades after a successful first therapy line, comprising high-dose therapy followed by autologous stem cell

transplantation. Notably, as LTS patients analyzed in this study received their first-line therapy more than a decade ago, the treatment regiments differ from those applied today, including a high amount of tandem-transplantation and the use of doxorubicin. We uncovered that MM long-term survivors display sustained immune alterations that are associated with the resurgence of the disease and correlated with disease activity. These disease-associated immune alterations are mediated by an inflammatory circuit driven by a tumor load-dependent infiltration of inflammatory T cells into the bone marrow. However, even in the absence of any measurable disease activity for years to decades, long-term alterations in the bone marrow ecosystem associated with defective immunity were observed.

Previous studies on immune reconstitution after exposure to cancer or cancer therapy, including autologous stem cell

**Fig. 6 | Immune remodeling in LTS patients is associated with future disease resurgence and defective immune function even in the absence of measurable disease. a** Correlation of malignant plasma cell fraction (CNV-based malignancy score) and degree of remodeling (mean DA-seq score). Spearman's Rho is indicated. **b** Correlation between mean CXCR3 expression in CD8 + T cells and CNV-based malignancy score; Spearman's Rho is indicated. **c** Correlation between the ratio of BM to PB CXCR3 + CD8 + T cells and BM cytological PC count. n(healthy) = 5, n(sust.CR) = 11, n(losingCR) = 11, n(ID) = 23. Spearman's Rho is indicated. **d** Comparison of BM to PB ratio of CXCR3 + CD8 + T cells between sust.CR patients (*n* = 11) and patients losing CR (*n* = 11). The dashed line indicates the mean ratio of healthy controls. Significance was tested by unpaired Wilcoxon rank sum test. **e** Bar plot summarizing fractions of dissimilarity-based classification by the clinical group for CD8 + T cells (healthy *n* = 3; complete remission (CR) *n* = 6; non-CR *n* = 5). **f** Similar to (**e**) but for classical monocytes. **g** Distribution of the dissimilarity score by clinical group within CD8 + T compartment. Large dots indicate sample means. **h** Similar to (**g**) but for classical monocytes. (**i**−**k**) Boxplots of indicated gene/genesets. The dashed line indicates the mean module score of

healthy control. Significance was tested by paired Wilcoxon rank-sum tests. **l** Fraction of CXCR3 + CD8 + T cells at ID, after induction, and high-dose melphalan and autologous stem cell transplantation (ASCT), *n* = 136 patients, significance determined by Wilcoxon signed-rank tests. Benjamini-Hochberg adjusted *p*-values are shown. **m** Correlation of CXCR3-positive T cells among paired samples from distinct therapy phases. Spearman's Rho is indicated. **n** Study design scheme; created with BioRender.com. **o** Intracellular cytokines in T cells of LTS patients (*n* = 17) and controls (*n* = 10). Significance was tested by unpaired Wilcoxon rank sum test. If not stated otherwise, paired BM samples from 11 MM patients at ID and LTS, and 3 healthy controls were used. Box plots: center line, median; box limits, first and third quartile; whiskers, smallest/largest value no further than 1.5*IQR from the corresponding hinge. Bar plots: Error bars indicate the standard error of the mean (SEM). Error bands in correlation graphs indicate a 95% confidence interval of best linear fit. All statistical tests were performed using a two-sided approach. Abbreviations: ID: initial diagnosis; LTS: long-term survival; CR: complete remission; BM: bone marrow; PB: peripheral blood.

transplantation, focused on the short-term impact. For example, te Boekhorst et al.[32]. and Schlenke et al.[33]. investigated the reconstitution of the T cell compartment in a mixed cohort of different hematological, as well as solid tumor patients. Both studies did not observe any signs of functional impairment in T cells from PB as measured by standard flow cytometry phenotyping. However, MM patients were underrepresented in both study cohorts. In a study focusing on short-term consequences of autologous SCT in MM, an impaired cytokine production of the T cell compartment was observed, concluding that the complete recovery of the immune system might require more time[34]. However, our study reveals long-term sustained molecular changes in the immune microenvironment, even in MM patients that were considered functionally cured, suggesting irreversible immunological scarring, as previously described in infectious diseases[6,7]. While our study focused on transcriptomic and immunological changes in LTS patients, a recent study identified clonal hematopoiesis as a common event upon long-term survival of pediatric cancers[28]. In a subset of Hodgkin Lymphoma survivors, therapy-related STAT3 mutations were detected that potentially also impact on T cell biology.

While our data support a non-genomic mechanism of sustained changes of the immune system in MM LTS, we cannot exclude that also genomic aberrations may contribute to some of the irreversible phenotypes we observed. Recent studies on MM have shown that exposure to high-dose melphalan increases the mutational burden by ∼10–20%. Nonetheless, the involvement of key myeloma driver genes is infrequent in most of these mutations[35]. Whether and how the immune system is affected by the mutagenic impact of melphalan still needs to be examined, desirably in a comparative study including patients treated with and without high-dose chemotherapy.

Our study suggests a tumor load-dependent inflammatory circuit in MM with the release of CXCL10 from myeloid cells causing the migration of CXCR3 + inflammatory T cells from the periphery to the BM, in line with previous reports in the context of cancer and vaccinations[36–38]. Inflammatory T cells and NK cells, in turn, act as major drivers for IFNG-mediated BM changes in a self-propelling circuit. This inflammatory circuit is initiated at ID and maintained in a subset of immune cells during LTS. Notably, immune remodeling in LTS patients was associated with future disease resurgence and impaired immune function even in the absence of measurable disease. However, it remains to be investigated whether the sustained proinflammatory bone marrow microenvironment in LTS patients actually promotes disease resurgence or contributes to disease control. Locally, such proinflammatory bone marrow microenvironments may serve as a basis for seeding and repopulation of circulating myeloma cells from extramedullary sites into the bone marrow. On the contrary, the recruitment of CXCR3-expressing T cells has been associated with an improved antitumor immunity, and IFNG and CXCL10 have been

shown to inhibit the growth of MM cells, pointing towards a potential role in immune-mediated tumor control[39,40].

Importantly, we demonstrate that disease-associated T cell trafficking can be used to track and reliably predict the re-initiation of the disease in the bone marrow of LTS patients by analyzing CXCR3 expression on CD8 + T cells in the peripheral blood. This highlights how disease associated changes in the microenvironment might be used in combination with MRD detection methods to predict resurgence of disease activity. While the detailed contributions of T cell migration to anti-cancer immunity remains to be investigated, targeting the introduced inflammatory circuit may offer potential avenues for new therapeutic strategies[41,42].

Of note, our study included paired samples of patients experiencing long-term remission after a single therapy line in the absence of any maintenance therapy for years. Due to continuous maintenance therapy as the new standard of care, this patient cohort is not recruitable nowadays and thus displays a highly valuable selection of patients to study the long-term consequences of cancer and cancer therapy in absence of potential biases associated with additional therapies.

Together, our study provides detailed insights into the molecular and cellular immune bone marrow ecosystem of MM long-term survivors, thereby revealing reversible and irreversible disease- and therapy-associated alterations of the immune compartment which can serve as diagnostic and predictive tools.

## Methods
### Human samples
**Ethics approval and consent to participate.** BM samples from healthy and diseased donors were obtained at Heidelberg University Hospital after informed written consent using ethic application numbers S-480/2011 and S-052/2022. BM aspirates were collected from the iliac crest. Healthy BM donors received financial compensation in some cases. For BM, mononuclear cells (BMMC) were isolated by Ficoll (GE Healthcare) density gradient centrifugation and stored in liquid nitrogen until further use. All experiments involving human samples were approved by the ethics committee of the Heidelberg University Hospital and were in accordance with the Declaration of Helsinki.

### Flow cytometry
**MRD analysis.** Flow cytometry for the detection of minimal residual disease (MRD) in fresh BM samples was performed according to the highly standardized flow cytometry approach developed and described by the Spanish Myeloma Collaborative Group using a commercially available EuroFlow 8-color 2-tube MM MRD Kit (Cytognos, Salamanca, Spain)[43]. Tube one contained multiepitope CD38-FITC,

CD56-PE (clone C5.9), CD45-PerCP-Cyanine5.5 (clone EO1), CD19-PE-Cyanine7 (clone 19-1), CD117-APC (clone 104D2) and CD81-APC-C750 (clone M38) antibodies. Tube two contained multiepitope CD38-FITC, CD56-PE (clone C5.9), CD45-PerCP-Cyanine5.5 (clone EO1), CD19-PE-Cyanine7 (clone 19-1), cytoplasmic polyclonal immunoglobulin (Ig) κ-APC goat and cytoplasmic polyclonal Igλ-APC-C750 antibodies. Drop-in CD27 Brilliant Violet 510 (clone O323, Biolegend, San Diego, USA) and CD138 Brilliant Violet 421 (clone MI15, BD, Heidelberg, Germany) antibodies were added to tubes one and two, according to manufacturer's instructions. Measurements were performed using BD FACSLyric (BD, Heidelberg, Germany) after the implementation of the EuroFlow Standard Operating Protocol for Instrument Setup and Compensation in FACSDiva (BD Biosciences, San Jose, CA, USA). Final data analysis was performed in Infinicyt 2.0 (Cytognos, Salamanca, Spain). An automated gating and identification tool (Cytognos, Salamanca, Spain) was used to support the identification of MM cells. Plasma cells were identified based on the co-expression of CD38 and CD138 antigens. An aberrant plasma cell expression profile was defined as CD45-low/negative, CD56-positive, CD19-negative, and light chain-restricted.

**Flow cytometry of cryopreserved BM samples.** Human BM samples were thawed in a water bath at 37 °C and transferred dropwise into RPMI-1640 10% FCS. Cells were centrifuged for 5 min at 350 × g and washed once with RPMI-1640 10% FCS. Cells were resuspended in FACS buffer (FB) (PBS 5% FCS 0.5 mM EDTA) co-incubated for 15 min at 4 °C. For analysis of CXCR3 expression on CD8 + T cells across different clinical groups, cells were stained with CD8-APC (1:30), CD3-APCR700 (1:50), CD45-APCH7 (1:20), CD4-FITC (1:20), CXCR3-PECy7 (1:20), CD194-BV421 (1:20), CD196-BV605 (1:20), CD152-PE (1:20) surface antibodies and FcR blocking reagent (Miltenyi). For analysis of CD69 expression on CD8 + T cells, cells were stained with CD8-BUV395 (1:50), CD4-BUV737 (1:50), CXCR4-BV421 (1:30), CD45RO-BV711 (1:50), CD69-FITC (1:10), CXCR3-PECy7 (1:20), CCR7-APC (1:50), CD3-APCCy7 (1:100) surface antibodies and FcR blocking reagent (Miltenyi). After washing with FB, all experiments were measured on BD FACSFortessa flow cytometer, equipped with 5 lasers, or BD FACSLyric flow Cytometer, equipped with three lasers.

**Single-cell RNA sequencing data**
**BM preparation, staining and sorting for gene expression analysis.** Human BM samples were thawed in a water bath at 37 °C and transferred dropwise into RPMI-1640 10% FCS. Cells were centrifuged for 5 min at 350 × g and washed once with RPMI-1640 10% FCS, followed by resuspension in FACS buffer (FB) (PBS 5% FCS 0.5 mM EDTA) containing CD45-PE (1:50) and CD3-APC (1:10) and FcR blocking reagent (Miltenyi) and incubation for 15 min at 4 °C. Cells were washed with FB. To exclude debris and ensure that actual cells were sorted for droplet-based scRNAseq, cells were stained with a DNA dye (Vybrant DyeCycle Violet, Thermo Fisher Scientific). For this purpose, 2.5 μl ml−1 Vybrant dye in cell suspension medium was incubated with 3 × 10^6 cells at 37 °C for 20 min in a water bath. Following the incubation, the cells were placed on ice and were sorted immediately for each experiment into 15 μl PBS containing 2% fetal bovine serum. For sorting of total BM cells, single, live cells were selected. For sorting of T cells, additionally, CD45 + CD3+ cells were gated (Supplementary Fig. 1e). Cells were sorted using a FACSAria Fusion or FACSAria II equipped with 100 μm nozzles, respectively. Sorted cell numbers were confirmed using a LUNA automated cell counter (Logos Biosystems). A volume of 33.8 μl of the cell suspension was used as input without further dilution or processing, with final concentrations of around 300 cells per μl.

**Single-cell RNA sequencing and data preprocessing.** Single-cell RNA sequencing libraries of BMMCs form healthy controls and MM patients were generated using 10x Genomics single-cell RNAseq

technology (Chromium Single Cell 3' Solution v2) according to the manufacturer's protocol and sequenced on an Illumina HiSeq4000 (paired end, 26 and 74 bp). Upon sequencing, FASTQ files were processed and aligned to the human reference genome GRCh38 (GEN-CODE v32) using the standard Cellranger pipeline (10x Genomics, v4.0).

**scRNA-seq data analysis**
All analyses were performed in R (v4.0.0). The output from the Cellranger pipeline was combined into one count matrix and further processed and analyzed using the Seurat framework (v4.0.1, (Hao et al. 2021)). Parameters are indicated when non-default settings for a specific function were used.

**Quality control of BM scRNA-seq data.** Cells were retained in the dataset if they had 200 – 40,000 UMIs, 400 – 6,000 features, and fewer than 10% mitochondrial reads. In addition, decontX()from the R package celda (v1.4.7[44],) was used to estimate and remove contaminating ambient RNA.

**Dimensionality reduction and clustering of BM scRNAseq-data.** Gene counts were log-normalized, and the top 2000 variable features were identified and scaled using default parameters of FindVariableFeatures() and ScaleData(). Dimensionality reduction of the scaled data was performed by principal component analysis (PCA). The top 50 PCs were then used to build a shared nearest neighbor graph (SNN, FindNeighbors(dims = 1:50)) for Louvain clustering (FindClusters(resolution = 0.7)) and uniform manifold approximation and projection (RunUMAP(Dims = 1:50)) of the data in two-dimensional space. Final cluster resolution and annotation was defined by evaluating known marker genes. Clusters with overlapping gene signatures were merged to reach overall cell-type resolution (*MetaClusters*). In order to achieve a more fine-granular filtering and annotation, each cell type (*MetaCluster*) was subsetted, and count matrices were separately processed again from variable feature selection and re-scaling to dimensionality reduction by PCA and subsequent clustering and UMAP representation. Clusters with contaminating gene expression profiles or aberrantly high mitochondrial and low housekeeping gene expression were considered as doublets, or low quality, respectively, and removed. Final cell annotation was then transferred back to the global BM count matrix. In addition, cells from patients treated with maintenance and induction therapy were removed.

**Copy number analysis.** Single-cell copy number analysis was performed using infercnv (v1.6.0, (Tickle T 2019)) with JAGS (v4.3.0, (Plummer 2003)). First, we generated a gene ordering file using a Python script provided by the infercnv developers (https://github.com/broadinstitute/infercnv/blob/master/scripts/gtf_to_position_file.py, 21 Apr 2021) and excluded all genes that were not part of this file. We only considered chromosomes 1-22 and, in order to avoid artefacts due to differential immunoglobulin gene expression, excluded all genes starting with "*IGH*", "*IGL*" or "*IGK*". The actual inferCNV analysis was performed separately for the plasma cells from each patient and utilized non-normalized decontX-corrected expression values. Plasma cells from the three healthy donors were used as reference cells. We disabled the filtering threshold regarding counts per cell and used the arguments "cutoff = 0.1", "cluster_by_groups = TRUE", "cluster_references = FALSE", "analysis_mode = 'subclusters'. "tumor_subcluster_pval = 0.05". "denoise = TRUE", "noise_logistic = TRUE", "HMM = TRUE", "HMM_type = 'i6'" and "num_threads = 1" within infercnv's function run(). Subsequently, we manually annotated the detected sub-populations as "healthy", "malignant" or "unclear" based on the denoised infercnv results. We additionally determined the major immunoglobulin light chain expressed by

malignant cells in a patient-wise fashion by inspecting the expression of the corresponding genes (*IGKC, IGLC1-7*). Afterward, we refined the malignancy annotation to reduce the number of cells that were wrongly classified as malignant. To this end, we compared immunoglobulin light chain gene expression (decontX-corrected and normalized) in each putatively malignant cell with the corresponding mean expression in its sub-population. If the expression of the patient-specific major light chain gene was less than half of the corresponding mean expression in the corresponding sub-population and the expression of another light chain gene was above 1.5 times the corresponding mean expression in the corresponding sub-population, a cell's classification was forced to "healthy". Copy number heatmaps were generated using ComplexHeatmap (v2.6.2[45]), circlize (v0.4.13[46],), scales (v1.1.1, (H. Wickham 2020)), magick (v2.7.3[47],) and imagemagick (v6.9.12[48],). Only cells from samples that were not obtained during induction and maintenance treatment are displayed.

**scRNA-seq quality control of T cell data.** Cells were kept in the dataset if they had between 500–20,000 UMIs, between 300–4000 detected features, and less than 10% mitochondrial reads. Clusters of contaminating cells, including myeloid cells, erythroid progenitors, and plasmablasts, were identified based on the expression of cell type-specific marker genes. Subsequently, decontX() from the R package celda (v1.4.7[44]) was applied on the count matrix to account for cross-contaminating reads using the contaminating cell types and remaining T cells as cluster labels. The final Seurat object was filtered to maintain only T cells, and the decontX matrix was used for all subsequent analyses.

**Classification of T cell subsets.** A reference dataset was generated from the T cell dataset by annotating cells based on the normalized decontX matrix (NormalizeData):

CD4: CD4 > 1.5 & CD8A == 0 & CD8B == 0 & TRDC == 0
CD8: (CD8A > 1.5 | CD8B > 1.5) & CD4 == 0 & TRDC == 0
gdT: TRDC > 1.5 & CD8A == 0 & CD8B == 0 & CD4 == 0

For each of these T cell subsets, dimensionality reduction was performed ((NormalizeData(), FindVariableFeatures(nfeatures = 1000), ScaleData(), RunPCA()), and cells were clustered to define the main cell states (FindNeighbours(reduction = 'pca',dims = 1:20), FindClusters(resolution = 0.4)). The subsets were then merged back into a combined reference dataset to annotate the complete T cell dataset with SingleR (v1.2.4[49],) taking "pruned.labels" output to split the T cell Seurat object into CD4, CD8, or gdT cell subsets for further analyses.

**CD8 subset analysis.** Dimensionality reduction and clustering was re-run (as above, except RunUMAP(dims = 1:20), FindClusters(resolution = 0.5)) as the final filtering step excluding a cluster-specific for cycling cells and then repeated to obtain a final version (as before, except FindClusters(resolution = 0.45)). Clusters were annotated to CD8 + T cell states based on the module score expression for custom gene signatures, which was added for each cell with AddModuleScore(): naive (genes: CCR7, TCF7, LEF1, SELL; cluster: (1), effector/central memory (genes: GPR183, CCR7, SELL, IL7R, CD27, CD28, GZMA, CCL5, S1PR1, GZMK, CXCR4, CXCR3, CD44; clusters: (2, 3, 5, 7), cytotoxic (genes: EOMES, TBX21, GZMB, PRF1, FASLG, GZMH, GZMA; cluster: (4). In addition, cluster 6 was annotated as KLRB1 + T cells based on the high expression level of the corresponding gene.

**CD4 subset analysis.** Similar to the CD8 + T cell dataset, cells were projected into a low dimensional space and grouped using graph-based clustering (as before, except FindClusters(resolution = 0.45)).

**Differential abundance analysis.** Changes in the composition of the BM microenvironment between the clinical states were evaluated by log2fold-change difference of each patient's cell type fraction from the corresponding healthy control's mean fraction. Prior to quantification of the compositions, plasma cells, and erythroid progenitors were excluded and evaluated separately. Further, plasma cells from patient P015 were excluded for cellular abundance analysis, as only the negative MACS fraction for plasma cell enrichment was available for single-cell RNA sequencing. Fractions and log2fold-change differences were tested for significance using an unpaired Wilcoxon rank sum test.

For cluster-independent differential abundance analysis, DA-seq was performed (Zhao et al. 2021[50]). The tool computes a multiscale score for each cell based on the k-nearest- neighborhood for k between 50 and 500. Following the multiscale score computation, a logistic regression classifier is trained to predict the enrichment state of each cell. Cells in differential abundant populations, whose neighborhoods are enriched with cells from one biological state, ID or healthy respectively, tend to be closer to each other in the score space. The logistic regression classifier is trained with L2 regularization to ensure smooth output and reduced outliers. The regularization hyperparameter lambda is optimized using cross-validation. As a result, a high value indicates that a cell is located in an enriched region, and low values indicate that a cell is in a depleted region. Cells with a DA measure > 0.95 and < − 0.95 were considered as differential abundant and were visualized on the UMAP. A continuous DA-seq score was calculated by subtracting scaled module scores (AddModuleScore()) for significantly up- and downregulated genes in differentially abundant cells.

**Dissimilarity analysis and aberrant cell classification.** To determine and quantify whether a cell is transcriptionally more similar to healthy cells or to perturbed counterparts in the disease state, we introduce a 'dissimilarity score'. It requires condition labels $i$ (in our case "Healthy" and "ID"), sample labels $j$ and a data matrix $X$. The analysis was performed per cell type to account for cell type-specific transcriptional differences. By default, we chose PCA coordinates of $n$ dimensions as dimension-reduced representations of our data, where $n$ was assessed by prior MetaCluster analysis. Cells were divided by condition and further sampled to adjust for equal group sizes. We computed the $k = 30$ nearest neighbors using the *FNN* package (v.1.1.3) to look at the condition distribution for each cell in the dataset. Dissimilarity was quantified by summing up the neighbors per condition with higher values meaning more neighboring cells from the diseased state (ID) as compared to healthy. To adjust for sampling effects, this process was iterated 100 times with changing seeds. Each cell is assigned the median dissimilarity and the final score is scaled between 0 and 1 between all conditions.

To allow group-wise comparisons between 'healthy-like' cells and most dissimilar, i.e., 'aberrant-like' cells among the clinical states, we used the automatic machine learning software H2O autoML (E. LeDell 2020). Initially, each cell was given a 'state' label ('healthy-like' or 'aberrant-like') based on the combination of the 'clinical state' ('Healthy' or 'ID') and the 'dissimilarity score'. The underlying 'dissimilarity score' threshold was defined as 99% of all cells from the healthy controls being labeled 'healthy-like', and applying this threshold on all patients' cells. Then, the top 500–1000 variable genes were computed for each cell population (see Supplementary Data 3) using *Seurat's* FindVariableFeatures(). To train and validate the models, training (80%) and test (20%) datasets were generated for each cell population using the createDataPartition() function from the *caret* package (v.6.0-91[51]). To have sufficient numbers of healthy plasma cells for model training and validation, healthy plasma cells from the 'Human Cell Atlas'[52] were integrated with our dataset applying the Scanorama algorithm with default parameters on all

features[53]. The partitioned datasets were then converted to H2O objects using the H2O library (H2O R version: 3.36.0.3. H2O cluster version: 3.36.0.3). The function h2o.automl() was used for the model training process using the train dataset and top n variable genes (500 or 1000) as input. Following parameters were set: max_models = 80 (which computes 82 models due to including the two Stacked Ensembles as default), max_runtime_secs_per_model = 7200, stopping_rounds = 5 and nfolds = 50 or nfolds = 5 (depending on dataset size). Moreover, a seed was set to ensure result reproducibility.

The top leader model (see Supplementary Data 3) was selected and used for label prediction on the respective test dataset. To assess label prediction accuracy for each model, a confusion matrix was generated, and the F1 score calculated using *caret's* confusionMatrix() function. The respective leader model was then used for classification and label prediction. After running h2o.predict(), additional filtering thresholds were applied (p0 > = 0.66 and p1 > = 0.66) on the internal probability values to differentiate between clearly defined (p0 > = 0.66 and p1 > = 0.66) and non-defined cells.

**Differential gene expression analysis.** Differential gene expression analyses were computed using a two-part generalized linear model implemented in MAST (v1.18.0[54],). The Hurdle model in MAST considers the bimodal expression distributions of single-cell data having either a strong gene expression or zero values (zero inflation). Normalized decontX corrected data of the whole human bone marrow or without the cells of ID were used as input. Genes with less than 10% expression across all libraries were filtered out. For the remaining genes, the hurdle model using the patients, the cell state, and CR status was fitted using the MAST function zlm(). The obtained coefficients for each variance-covariance and gene were reported with summary().

**Gene set enrichment analysis.** Gmt files containing gene set collections were obtained from the Molecular Signatures Database (c2.cp.v7.4.symbols.gmt, c5.all.v7.4.symbols.gmt, h.all.v7.4.symbols.gmt[55,56],). To search for enriched terms of cells from patients at initial diagnosis being classified as 'aberrant' compared to 'healthy' cells from healthy donors, their average log2 fold-change among all genes was calculated. Subsequently, genes were sorted by their average log2 fold-change and used for multilevel GSEA with the fgsea R package (v1.14.0[57],). Results were filtered for padj < 0.05 and sorted by their normalized enrichment score (NES). Significantly enriched gene sets of interest were further evaluated by calculating a module score for the corresponding gene signature, or for specified leading-edge genes in each cell using AddModuleScore()in Seurat and comparing these modules in cell types of interest between the clinical groups.

To systematically assess enriched gene sets between the clinical groups including the 'complete remission' status, all gene set collections were combined into one gene matrix transposed file (gmt) as input for GSEA, which was then performed as stated above. Top 100 enriched (NES) and significant (*p* < 0.05) scores were selected per corresponding cell type, translated into a ModuleScore, and tested for significance between the clinical and CR states using a paired Wilcoxon signed rank test.

**GO overrepresentation analysis.** To identify enriched terms among the DEGs from the MAST analyses, GO overrepresentation analysis was performed with the clusterProfiler R package (v3.16.1[58],). The function enricher() was used to run GO analysis based on the same gmt files as used for GSEA.

**Surfaceome filtering.** DEGs from the MAST comparison of aberrant-like cells from patients at initial diagnosis against healthy-like cells from healthy donors within the memory CD8 + T cell subset were filtered for surface proteins using Cell Surface Protein Atlas data,

including validated surfaceome proteins[59]. Briefly, surface proteins annotated in Table A of the file http://wlab.ethz.ch/cspa/data/S2_File.xlsx (21 Apr 2021) were filtered for the category '1 - high confidence' and DEGs were filtered for the intersection with the remaining gene symbols in the surfaceome table.

**Cell-cell interaction analysis.** Cell-cell interactions were inferred with CellphoneDB2.0[24] using normalized and decontX corrected count data of the human bone marrow data set. Receptor-ligand interactions were inferred for mean expression within each cell label cluster as well as for clusters having the combined information of cell label and DA-Seq information. For downstream analyses, significant interactions with an adjusted *p*-value < 0.05 were considered, which required an expression of receptor and ligand in at least 10% of the cells per cluster. CellphoneDB2.0 was computed per patient, and the significant interaction counts were grouped over the respected disease subgroups.

**RNA Velocity.** To investigate developmental dynamics, scVelo (v0.2.4[60],) in combination with Velocyto (v0.17.17[61],) in Python (v3.9.7) was used. Reads were annotated as spliced, unspliced, and ambiguous. The pipeline was run individually for each sample and data from resulting loom files were combined. Cells were subsetted based on prior analysis of CD8 + T cells. Splicing kinetics were recovered using recover_dynamics() with standard parameters, velocities were computed using velocity (mode = 'dynamical') and the velocity graph was calculated by velocity_graph() with standard parameters. Finally, for visualization, summarized velocity vectors are plotted using the velocity_embedding_stream() function in UMAP space in combination with the dissimilarity score. For plotting of single marker expression, velocity() was used.

## Bulk RNA-sequencing and TCR clonotyping

**BM preparation, staining, and sorting.** For sorting of CXCR3 + and CXCR3- cells, BM and PB samples of MM patients were thawed and processed as described above. Cells were stained with CD3-APCCy7 (1:100), CD4-BUV737 (1:50), CD8-BUV395 (1:50) and CXCR3-PECy7 (1:20) antibody. For sorting of CXCR3 + and CXCR3- cells, single, live CD3 + CD4-CD8+ cells were gated and sorted as CXCR3- or CXCR3 + cells, respectively. 1000 CXCR3 + and CXCR3-CD8 + T cells from each sample were sorted on FACSAria Fusion equipped with a 100 μm nozzle.

**Bulk RNA-sequencing and gene expression analysis.** RNA was isolated using the PicoPure RNA Isolation Kit (ThermoFisher), bulk RNA-sequencing libraries were generated using the SMART Seq Stranded Total RNA-Seq kit (Takara) and sequenced using the Illumina NovaSeq 6000 platform (2 × 100 bp). Adapter trimming was performed with Skewer (v0.2.2[62],). Reads were aligned to human reference GRCh38 using STAR (v2.5.2b[63],) and gene count tables were generated using Gencode v.32 annotations. Differential expression between samples was tested using the R/Bioconductor package DESeq2 (v1.30.1[64],). Sample origin (BM vs. PB) was added to the design formula (condition: CXCR3 + vs. CXCR3- CD8 + T) to retrieve significantly upregulated genes for CXCR3 + CD8 + T cells within the BM (termed bulkRNA Remodeling Module).

**TCR clonotype analysis.** Analysis and quantification of the TCR receptor profiles, statistical analysis, and visualization were performed using three main tools: MiXCR (v3.0.13[65],), VDJtools (v1.2.1[66],) and immunarch (v0.6.6, (ImmunoMindTeam 2019)). Raw bulk RNA sequencing data of sorted CD8 + T cells in FASTQ format was used as the input for the TCR clonotype analysis. Analyze shotgun command of MiXCR was used to align variable (V), diversity (D), joining (J), and constant (C) genes of T-cell receptors, correct PCR and sequencing errors, assemble bulk RNA-seq reads by CDR3 region to the reference

IMGT[67] library and export bulk TCR clonotypes. To this end, the default parameters recommended by the developers for RNA-seq data were used. Basic analysis, diversity estimation and repertoire overlap analysis modules of VDJtools were then used for the downstream analysis of the bulk TCR clonotypes provided by the MiXCR output. For the TCR repertoire clonality comparison between groups, the clonality metric was calculated as [1 – normalized Shannon Wiener Diversity Index]. Significant differences were evaluated by paired Wilcoxon signed rank test. For TCR repertoire overlap quantification, the Jaccard index was utilized. Hierarchical clustering of the quantified TCR repertoire overlap was then performed using the hclust() function of the R MASS package. JoinSamples() command of VDJtools were used to highlight overlapping TCR clonotypes by representative CDR3 amino acid sequence between CXCR3 status and sample origin (BM, PB) of CD8 + T cells of single patients. In addition, the frequency of the top 10 most abundant TCR clonotypes across different samples was tracked using immunarch. Clonotype tracking was performed by the representative CDR3 amino acid sequence of TCR clonotypes.

### T cells in vitro cytokine assay
CD3 + T cells were enriched from the BM of 30 MM patients using the Pan T cell isolation kit with MS columns (both Miltenyi Biotec, Bergisch, Germany). $5 \times 10^5$ CD3 + T cells were plated in 0.5 ml T cell expansion medium (Stemcell Technologies, Cologne, Germany) with 50 IE/ml IL-2, 1 % Pen/Strep (both from Sigma-Aldrich, Taufkirchen, German) in 24 well-plates and incubated overnight at 37 °C, 5 % CO₂. On the next day, GolgiStop (0.66 µl/ml) (both BD Biosciences, Heidelberg, Germany) was added, and cells were stimulated with PMA (50 ng/ml) and Ionomycin (1 µg/ml) (both Sigma-Aldrich, Taufkirchen, Germany). 6 h after incubation, intracellular staining was performed using transcription factor buffer set (BD Biosciences, Heidelberg, Germany) according to manufacturer's instructions. Briefly, cells were washed twice in PBS and stained with cell surface antibodies CD3-APCCy7 (1:100), CCR7-APC (1:50), CD45RO-BV711 (1:50), CD4-BUV737 (1:50) and CD8- BUV395 (1:50) for 20 min at 4 °C. Subsequently, antibody-conjugated cells were fixed and permeabilized for intracellular staining before washed twice with 1x Perm/Wash buffer and stained with antibodies against intra-cellular markers (TNFa-PECy7 (1:100), IFNG-PE (1:50) and Il-2-BV421 (1:50)) at 4 °C for 45 min. Cells were washed twice with 1x Perm/Wash buffer, and measurements were acquired on cell analyzer FACS Lyrics (BD Bioscience, Heidelberg, Germany). Controls without PMA and Ionomycin stimulation were included in this assay. Flow cytometry data were visualized in FlowJo (BD Bioscience, Heidelberg, Germany).

### CHIP genotyping
**NGS sample preparation and library generation.** Genomic DNA was isolated using the QIAamp DNA Mini or Micro Kit (Qiagen) according to the manufacturer's instructions. Input DNA quantitation was performed using a Quantus fluorometer (Promega) with 50 ng input per sample. DNA was then processed using the TruSight Myeloid Sequencing Panel Kit (Illumina), which includes the hybridization and extension-ligation of oligos, followed by PCR amplification with specific dual-index primers and adapters. The quality and size of PCR products were examined via agarose gel electrophoresis. After PCR clean-up via AMPure XP beads, the normalized libraries were pooled and loaded on a MiniSeq for sequencing using the MiniSeq High Output Kit (300 cycles) to generate 2 × 150 read lengths. The TruSight Myeloid Sequencing Panel comprises 568 amplicons interrogating 54 genes associated with myeloid neoplasms: ABL1, ASXL1, ATRX, BCOR, BCORL1, BRAF, CALR, CBL, CBLB, CBLC, CDKN2A, CEBPA, CSF3R, CUX1, DNMT3A, ETV6, EZH2, FBXW7, FLT3, GATA1, GATA2, GNAS, HRAS, IDH1, IDH2, IKZF1, JAK2, JAK3, KDM6A, KIT, KMT2A, KRAS, MPL, MYD88, NOTCH1, NPM1, NRAS, PDGFRA, PHF6, PTEN, PTPN11, RAD21,

RUNX1, SETBP1, SF3B1, SMC1A, SMC3, SRSF2, STAG2, TET2, TP53, U2AF1, WT1 and ZRSR2.

**Bioinformatics analysis.** Data alignment was performed with Local Run Manager v2.0.0 (Illumina), aligner BWA, and Quality Control parameters were examined via Sequencing Analysis Viewer v2.4.7 (Illumina). For data annotation, Variant Studio v3.0 (Illumina) was used, and variants were interpreted from a threshold of read depth ≥ 500 and allelic frequency ≥ 2%. Interpretation and classification of variants is based on pathogenicity, defined by varsome (https://varsome.com), localization in functional domains, examined by cBioPortal for Cancer Genomics (https://www.cbioportal.org) and availability of a COSMIC-ID (Catalog Of Somatic Mutations In Cancer - https://cancer.sanger.ac.uk/cosmic) as well as publications of the corresponding variant.

### Multiplex Immunofluorescence
The frequency, localization, and spatial proximity of T cell subpopulations and plasma cells, as well as their expression of respective markers LAT1 and CXCR3, was analyzed by multispectral imaging (MSI). Formalin-fixed and paraffin-embedded (FFPE) bone marrow (BM) biopsies of patients with MM (n = 33), and control BM tissue of patients with B cell Non-Hodgkin lymphomas without evidence for BM infiltration (n = 12) and myelodysplastic syndromes were collected between 2017 and 2020 at the Institute of Pathology of the Medical Faculty of the Martin-Luther University Halle-Wittenberg, Germany. The use of FFPE tissue samples was approved by the Ethical Committee of the Medical Faculty of the Martin Luther University Halle-Wittenberg, Halle, Germany (2017-81). The staining procedure was performed as recently described (Bauer et al., 2020). The marker panel used for staining included monoclonal antibodies (mAb) directed against CD3 (Labvision. Germany. clone SP7), CD8 (Abcam, Cambridge, UK, clone SP16), MUM1 (Dako, USA, clone MUM1p), LAT1 (Abcam, Cambridge, UK, clone EPR17573) and CXCR3 (Abcam, Cambridge, UK, clone ab133420). Briefly, all primary mAb were incubated for 30 min. Tyramide signal amplification (TSA) visualization was performed using the Opal seven-color IHC kit containing fluorophores Opal 520, Opal 540, Opal 570, Opal 620, and Opal 690 (Perkin Elmer Inc., Waltham, MA, USA), and DAPI. Stained slides were imaged employing the PerkinElmer Vectra Polaris platform. Cell segmentation and phenotyping of the cell subpopulations were performed using the inForm software (PerkinElmer Inc., USA). The frequency of all immune cell populations analyzed, and the cartographic coordinates of each stained cell type were obtained. The statistical analysis of the cell populations' spatial distribution was performed in R.

### Reporting summary
Further information on research design is available in the Nature Portfolio Reporting Summary linked to this article.

## Data availability
The scRNAseq and bulk RNAseq data generated in this study have been deposited in the European Genome-Phenome Archive (EGA) under accession code EGAS00001006980 https://ega-archive.org/search/EGAS00001006980. The processed scRNAseq data are available at Figshare https://doi.org/10.6084/m9.figshare.26935744. Source data are provided with this paper.

## Code availability
The source code used for generating the figures and the underlying data supporting the findings of this study have been deposited in Figshare and are publicly available under https://doi.org/10.6084/m9.figshare.26935744.

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

## Acknowledgements

We would like to thank members of the Brors, Haas, and Hundemer laboratories for helpful discussions. Moreover, we thank members of the DKFZ Single cell open lab and the DKFZ flow cytometry for support. We would like to thank A. Mahmoud for an initial analysis of the single-cell RNA-seq dataset as part of his PhD thesis. This study was supported by a grant from the Black Swan Research Initiative of the International Myeloma Foundation, the German Bundesministerium für Bildung und Forschung (BMBF) through the Juniorverbund in der Systemmedizin 'LeukoSyStem' (FKZ 01ZX1911D), the German Research Foundation (DFG) through the project HA 8790/3-1 and the Dietmar Hopp Foundation. The contribution by R.L. was supported by the DKFZ Clinician Scientist Program funded by the Dieter Morszeck Foundation.

## Author contributions

R.L., S.H., M.H., and H.G. conceived the study. R.L. performed the single-cell RNA sequencing experiments with the help from T.B. and S.H. R.L., M.B., and M.A. performed the experimental validations and functional experiments with help from M.H., D.V., N.P., M.A.B., P.S., F.G., and C.W. F.G. and M.S. conducted the majority of bioinformatics analyses with conceptional input from R.L., S.H., M.H., B.B., and C.I. L.J.-S., M.B., S.Y., N.B., L.S.B., B.A., S.S., A.S., C.I., and G.S. performed additional bioinformatics analyses. S.H. and M.H. supervised the experimental work. C.I. and B.B. supervised the bioinformatics analyses with help and conceptional input from S.H., K.R., and D.H. D.V., T.B., D.H., B.D., N.W., M.S.R, C.W, A.T., H.G., and C.M.-T. provided clinical samples and conceptional input on data interpretation. R.L., S.H., F.G., M.S., M.H., N.B., L.J.-S., S.Y, M.B., A.S., and C.I. wrote the manuscript and prepared figures. All authors have carefully read the manuscript.

## Funding

## Competing interests

C.M.-T. received research funding from Bioline RX and Pfizer. M.B. has served as a consultant for Takeda and Novartis; received research funding from Novartis and travel support from Celgene, Amgen, and Janssen. H.G. received funding from the International Myeloma Foundation – Black Swan and the Dietmar-Hopp-Foundation. The other authors declare no competing interests related to this study.

## Additional information

Raphael Lutz[1,2,3,4,30], Florian Grünschläger[3,4,5,30], Malte Simon[5,6,7,30], Mohamed H. S. Awwad [1], Marcus Bauer[8], Schayan Yousefian [9,10,11], Niklas Beumer [5,6,12,13,14], Lea Jopp-Saile[3,4,5,10], Anastasia Sedlmeier [15], Llorenç Solé-Boldo[9,10,11], Bogdan Avanesyan [9,10,11], Dominik Vonficht[3,4,5], Patrick Stelmach [3,4], Georg Steinbuss[1], Tobias Boch[3,4,16], Simon Steiger [17], Marc-Andrea Baertsch[1,18], Nina Prokoph[1,18], Karsten Rippe [17], Brian G. M. Durie[19], Claudia Wickenhauser[8], Andreas Trumpp [3,4], Carsten Müller-Tidow [1,20], Daniel Hübschmann [3,15,21], Niels Weinhold [1,18], Marc S. Raab [1,18], Benedikt Brors [6,22,23,24,31] ✉, Hartmut Goldschmidt [25,31] ✉, Charles D. Imbusch [6,26,27,28,31] ✉, Michael Hundemer [1,31] ✉ & Simon Haas [3,4,9,10,11,29,31] ✉

[1]Department of Medicine V, Hematology, Oncology and Rheumatology, Heidelberg University Hospital, Heidelberg, Germany. [2]Oncology Center Speyer, Speyer, Germany. [3]Heidelberg Institute for Stem Cell Technology and Experimental Medicine (HI-STEM gGmbH), Heidelberg, Germany. [4]Division of Stem Cells and Cancer, German Cancer Research Center (DKFZ) and DKFZ–ZMBH Alliance, Heidelberg, Germany. [5]Faculty of Biosciences, Heidelberg University, Heidelberg, Germany. [6]Division of Applied Bioinformatics, German Cancer Research Center (DKFZ), Heidelberg, Germany. [7]Leibniz Institute for Immunotherapy (LIT), Regensburg, Germany. [8]Institute of Pathology, University Hospital Halle, Martin Luther University Halle-, Wittenberg, Germany. [9]Berlin Institute of Health (BIH) at Charité Universitätsmedizin, Berlin, Germany. [10]Berlin Institute for Medical Systems Biology, Max Delbrück Center for Molecular Medicine in the Helmholtz Association, Berlin, Germany. [11]Charité Universitätsmedizin, Berlin, Germany. [12]DKFZ-Hector Cancer Institute at the University Medical Center Mannheim, Mannheim, Germany. [13]Department of Personalized Oncology, University Hospital Mannheim, Medical Faculty Mannheim, University of Heidelberg, Mannheim, Germany. [14]Division of Personalized Medical Oncology (A420), German Cancer Research Center (DKFZ), Heidelberg, Germany. [15]Computational Oncology, Molecular Precision Oncology Program, National Center for Tumor Diseases (NCT) Heidelberg and German Cancer Research Center (DKFZ), Heidelberg, Germany. [16]Department of Hematology and Oncology, University Hospital Mannheim, Mannheim, Germany. [17]Division of Chromatin Networks, German Cancer Research Center (DKFZ) and BioQuant, Heidelberg, Germany. [18]CCU Molecular Hematology/Oncology, German Cancer Research Center (DKFZ), Heidelberg, Germany. [19]Cedars-Sinai Medical Center, Los Angeles, CA, USA. [20]Molecular Medicine Partnership Unit EMBL and University Hospital Heidelberg, Heidelberg, Germany. [21]Innovation and Service Unit for Bioinformatics and Precision Medicine (BPM), German Cancer Research Center (DKFZ), Heidelberg, Germany. [22]Medical Faculty and Faculty of Biosciences, Heidelberg University, Heidelberg, Germany. [23]National Center for Tumor Diseases (NCT), Heidelberg, Germany. [24]German Cancer Consortium (DKTK), Core Center Heidelberg, Heidelberg, Germany. [25]Department of Medicine V, Hematology, Oncology and Rheumatology, GMMG Studygroup, Heidelberg University Hospital, Heidelberg, Germany. [26]Institute of Immunology, University Medical Center Mainz, Mainz, Germany. [27]Research Center for Immunotherapy, University Medical Center Mainz, Mainz, Germany. [28]German Cancer Consortium (DKTK), Partner Site Frankfurt/Mainz, Mainz, Germany. [29]Precision Healthcare University Research Institute, Queen Mary University of London, London, UK. [30]These authors contributed equally: Raphael Lutz, Florian Grünschläger, Malte Simon. [31]These authors jointly supervised this work: Benedikt Brors, Hartmut Goldschmidt, Charles D. Imbusch, Michael Hundemer, Simon Haas. ✉e-mail: b.brors@dkfz-heidelberg.de; hartmut.goldschmidt@med.uni-heidelberg.de; c.imbusch@dkfz-heidelberg.de; michael.hundemer@med.uni-heidelberg.de; simon.haas@bih-charite.de

