## [Peer Review File · Nature Communications]

Multiple myeloma long-term survivors exhibit sustained immune alterations decades after first-line therapyEditorial Note: This manuscript has been previously reviewed at another journal. This document only contains reviewer comments and rebuttal letters for versions considered at *Nature Communications*.

REVIEWER COMMENTS

Reviewer #1 (Remarks to the Author):

The authors provided additional analyses and novel interpretation of their data. But I still feel the small sample size, also of the healthy controls and the enormous heterogeneity of the patients and their disease (standard risk vs high risk, one vs two transplants, different maintenance strategies that very likely have an impact on the tumor microenvironment in spite of being stopped some years ago and finally, the presence or lack of malignant plasma cells in the samples – without care biopsies or imaging to monitor for residual cells outside of the site of puncture does not support the conclusions drawn from this analysis.

Reviewer #2 (Remarks to the Author):

This study by Lutz et al. aims to investigate alterations of the bone marrow immune microenvironment in patients with long-term remissions after a first line treatment for multiple myeloma. Based on the results of this study, the authors conclude that multiple myeloma long-term survivors display pronounced immune alterations driven by an inflammatory immune circuit characterized by CXCR3-CXCL9/10/11 axis.

In this revised edition, the authors have conducted a substantial amount of experiments and data evaluation in response to the review comments, especially addressing the major concern from both reviewers about the effect of intensive therapy, including ASCT. Considering the originality of the research topic and the interesting results, it will appeal to the readers of *Nature Communications*.

Minor points:

(1) L433–441: the authors discuss the expression level of CXCR3 in the main text. However, the figures corresponding these sentences discuss the proportion of CXCR3 expressing cells. I would recommend revising the main text accordingly.

(2) Figure 6l: the single ASCT and tandem ASCT are grouped together. However, as shown in Response Figure 2f, it would be more appropriate to separate them. Please consider making this change.

Point-by-point reply – second revisions

Reviewer #1 (Remarks to the Author):

The authors provided additional analyses and novel interpretation of their data. But I still feel the small sample size, also of the healthy controls and the enormous heterogeneity of the patients and their disease (standard risk vs high risk, one vs two transplants, different maintenance strategies that very likely have an impact on the tumor microenvironment in spite of being stopped some years ago and finally, the presence or lack of malignant plasma cells in the samples – without care biopsies or imaging to monitor for residual cells outside of the site of puncture does not support the conclusions drawn from this analysis.

We thank this reviewer for the comments, remarks and suggestions from the original and current round of revisions, which we have broadly addressed with a wide array of analyses and experiments. In acknowledgement to the reviewer statement, we have further toned down any causality statements, as the presented study is mostly associative by nature. All statements drawn in the revised version of the manuscript are fully supported by the presented experimental and analytical data.

Reviewer #2 (Remarks to the Author):

This study by Lutz et al. aims to investigate alterations of the bone marrow immune microenvironment in patients with long-term remissions after a first line treatment for multiple myeloma. Based on the results of this study, the authors conclude that multiple myeloma long-term survivors display pronounced immune alterations driven by an inflammatory immune circuit characterized by XXCR3-CXCL9/10/11 axis.

In this revised edition, the authors have conducted a substantial amount of experiments and data evaluation in response to the review comments, especially addressing the major concern from both reviewers about the effect of intensive therapy, including ASCT. Considering the originality of the research topic and the interesting results, it will appeal to the readers of Nature Communications.

We like to thank this reviewer for the constructive feedback that substantially improved our study. We have now incorporated all additionally enquired points.

Minor points:

(1) L433–441: the authors discuss the expression level of CXCR3 in the main text. However, the figures corresponding these sentences discuss the proportion of CXCR3 expressing cells. I would recommend revising the main text accordingly.

We thank the reviewer for raising this point. We have now revised the main text accordingly.

(2) Figure 6l: the single ASCT and tandem ASCT are grouped together. However, as shown in Response Figure 2f, it would be more appropriate to separate them. Please consider making this change.

We agree with the reviewer and have incorporated an updated version of Figure 6I, separately illustrating single and tandem ASCT.